# Overcoming the cytoplasmic retention of GDOWN1 modulates global transcription and facilitates stress adaptation

**Zhanwu Zhu[1], Jingjing Liu[1], Huan Feng[1], Yanning Zhang[1], Ruiqi Huang[2], Qiaochu Pan[2], Jing Nan[1], Ruidong Miao[1], Bo Cheng[1,3]***

[1]School of Life Sciences, Lanzhou University, Lanzhou, China; [2]Cuiying Honors College, Lanzhou University, Lanzhou, China; [3]Key Laboratory of Cell Activities and Stress Adaptations, Ministry of Education, Lanzhou University, Lanzhou, China

**\*For correspondence:**
bocheng@lzu.edu.cn

**Competing interest:** The authors declare that no competing interests exist.

**Abstract** Dynamic regulation of transcription is crucial for the cellular responses to various environmental or developmental cues. Gdown1 is a ubiquitously expressed, RNA polymerase II (Pol II) interacting protein, essential for the embryonic development of metazoan. It tightly binds Pol II *in vitro* and competitively blocks the binding of TFIIF and possibly other transcriptional regulatory factors, yet its cellular functions and regulatory circuits remain unclear. Here, we show that human GDOWN1 strictly localizes in the cytoplasm of various types of somatic cells and exhibits a potent resistance to the imposed driving force for its nuclear localization. Combined with the genetic and microscope-based approaches, two types of the functionally coupled and evolutionarily conserved localization regulatory motifs are identified, including the CRM1-dependent nucleus export signal (NES) and a novel Cytoplasmic Anchoring Signal (CAS) that mediates its retention outside of the nuclear pore complexes (NPC). Mutagenesis of CAS alleviates GDOWN1's cytoplasmic retention, thus unlocks its nucleocytoplasmic shuttling properties, and the increased nuclear import and accumulation of GDOWN1 results in a drastic reduction of both Pol II and its associated global transcription levels. Importantly, the nuclear translocation of GDOWN1 occurs in response to the oxidative stresses, and the ablation of *GDOWN1* significantly weakens the cellular tolerance. Collectively, our work uncovers the molecular basis of GDOWN1's subcellular localization and a novel cellular strategy of modulating global transcription and stress-adaptation via controlling the nuclear translocation of GDOWN1.

## Editor's evaluation

This important study identifies two distinct nuclear export elements and a strong cytoplasmic anchoring sequence in the GDOWN1 transcription factor that restricts its nuclear import and its ability to inhibit RNA polymerase II transcription. The study shows how this mechanism is modulated in stress conditions that promote GDOWN1 nuclear localization as part of a protective response. This study presents compelling evidence for the role of GDOWN1 in transcriptional regulation and should be of wide general interest.

## Introduction

In eukaryotes, RNA Polymerase II (Pol II) catalyzes the RNA synthesis of the clear majority of protein coding genes and a great number of non-coding genes in eukaryotic genomes (*Haberle and Stark, 2018*; *Osman and Cramer, 2020*). The Pol II-associated transcription machinery is composed of the 12-subunit Pol II and a collection of dynamically bound and delicately coordinated factors, including

general transcription factors (TFIID, TFIIA, TFIIB, TFIIF, TFIIE, and TFIIH)(*Cramer, 2019*; *Fischer et al., 2019*), Pol II processivity-controlling factors (such as the writer, reader, and eraser factors for modifying and recognizing the carboxyl terminal domain [CTD] of RBP1 (*Hsin and Manley, 2012*; *Jeronimo et al., 2016*; *Sansó and Fisher, 2013*; *Yurko and Manley, 2018*), the positive or negative elongation factors and the termination factors) (*Core and Adelman, 2019*; *Jonkers and Lis, 2015*; *Proudfoot, 2016*; *Zhou et al., 2012*), and the co-transcriptional RNA processing and modifying factors (such as the capping enzymes, splicing machinery, RNA modification enzymes, and RNA cleavage factors) (*Kachaev et al., 2020*; *Kilchert and Vasiljeva, 2013*; *Neugebauer, 2019*; *Noe Gonzalez et al., 2021*; *Schier and Taatjes, 2020*; *Sun et al., 2020*). Many of these Pol II-binding and regulatory factors are not only essential for facilitating the production and processing of transcripts but also play critical roles in dynamic integrating the intracellular and extracellular information and adjusting Pol II's target specificity and enzymatic activities in real time to maintain cell homeostasis (*Lynch et al., 2018*; *McNamara et al., 2016*; *Muniz et al., 2021*; *Schier and Taatjes, 2020*).

Gdown1 was initially identified as a protein copurified with Pol II in calf thymus and porcine liver, and designated as the 13th subunit of Pol II due to its high-affinity interaction with Pol II (*Hu et al., 2006*). Data from *in vitro* transcription assays along with EMSA and chemical crosslinking-based structural analyses have demonstrated that Gdown1 interacts with multiple Pol II subunits via its various regions and strongly inhibits the binding and/or functions of a series of transcription regulatory factors, including the factors required for transcription initiation, such as TFIIF (*Jishage et al., 2012*; *Jishage et al., 2018*), the factors involved in the productive elongation such as RTF1/PAF1C (*Ball et al., 2022*), and the transcription termination factor such as TTF2 (*Cheng et al., 2012*).

Although the biochemical properties support Gdown1's potential in regulating Pol II transcription, it is largely unknown how exactly its regulatory activities are executed under the physiological circumstances. Knockout (*KO*) of *Gdown1* in flies and mice caused embryonic lethality, and moreover, the attempt of establishing a *Gdown1*$^{-/-}$ mouse embryonic stem cell line failed, pointing out its essential roles during the embryonic development (*Jishage et al., 2020*; *Jishage and Roeder, 2020*; *Jishage et al., 2018*). Interestingly, Gdown1 has been reported as a nucleocytoplasmic shuttling protein in flies. It colocalizes with Pol II in the nuclei at the transcriptionally silent syncytial blastoderm stage and moves to the cytoplasm at the later blastoderm stage when the global transcription is activated, and meanwhile, Gdown1 is found to be retained in the nuclei of the transcriptionally silent pole cells (*Jishage and Roeder, 2020*). Although not being proven yet, these findings imply that controlling the nucleocytoplasmic shuttling of Gdown1 may be tightly associated with tuning the transcriptional regulatory activities. At the beginning of the embryonic development, the nuclear Gdown1 may serve as a global transcriptional inhibitor, and once the embryos get adequately prepared, exclusion of Gdown1 from the nucleus provides an effective way to promote zygotic genome activation (ZGA). Further studies are certainly required to explore the functional and regulatory mechanisms behind and find out whether the above phenomena revealed in flies are similarly applied in higher animals or in other situations.

Evidence is piling up that Gdown1 also plays critical roles in somatic cells. It is expressed throughout the whole life cycle of flies (*Jishage et al., 2018*) and ubiquitously present across various types of mouse tissues (data not shown). Mice with Gdown1 specifically knocked out in liver were found viable and relatively normal, yet tended to trigger the quiescent hepatocytes to re-enter cell cycle in the absence of hepatic injury, highlighting its important roles in maintaining the homeostasis of hepatocytes (*Jishage et al., 2020*). Further ChIP-Seq analyses revealed that Gdown1 bound with the elongating Pol II at many genes in liver (*Jishage et al., 2020*). However, the RNA-Seq results unexpectedly revealed that the expression levels of those direct targets of Gdown1 were dramatically reduced upon its ablation, suggesting a positive effect of Gdown1 on transcription (*Jishage et al., 2020*). Thus, it is necessary to clarify the underlying reasons behind the apparently opposite transcriptional regulatory effects of Gdown1 observed in somatic cells and in the defined *in vitro* transcription assays.

To further explore GDOWN1's functions in somatic cells, we started out by examining its subcellular localization in many cultured human cell lines and confirmed that it was predominantly localized in the cytoplasm. Based on the known functions, it's reasonable to presume that human GDOWN1 is a nucleocytoplasmic shuttling protein. However, our data demonstrated that GDOWN1 was subjected to a very tight restriction for its nuclear import under the regular cell culture conditions. Based on these findings, we established various mutagenesis-based screening assays of GDOWN1 and identified

multiple intrinsic localization-regulatory signals and their working mechanisms. In addition, manipulation of GDOWN1's nuclear translocation caused a significant reduction of both the protein levels of Pol II and the global transcription. Its massive and constant nuclear accumulation resulted in a severe growth inhibition and even triggered cell death. Moreover, we provided the experimental evidence that the nuclear import of GDOWN1 was naturally induced upon certain cellular stresses and its genetic ablation resulted in a decreased cell viability under stresses. Overall, our data revealed a novel function of GDOWN1 in facilitating stress adaptation via modulating transcriptional homeostasis, and this protective strategy was achieved by its stress-responsive nuclear translocation.

## Results

### Gdown1 is primarily a cytoplasmic localized protein in mammalian somatic cells

We started out to detect the subcellular localization of GDOWN1 in the cultured human cell lines by ectopically expressing GDOWN1 fused with a fluorescent tag at its N- or C-terminus or simply with a Flag tag. Consistent with the previous observation in adult flies (*Jishage et al., 2018*) and the recent report in human somatic cells (*Ball et al., 2022*), the localization signals of GDOWN1 were exclusively present in the cytoplasm of HeLa cells, regardless of the position or size of the fused tags (*Figure 1A*). To explore GDOWN1's functions in the nucleus, two nuclear localization signals (NLS) were fused to GDOWN1 at each end, which were known to be efficient for driving the 160 kDa SpCas9 protein into the nucleus in a commonly used CRISPR-vector pX459 (*Ran et al., 2013*). Unexpectedly, the addition of two NLS did not affect GDOWN1's cytoplasmic localization (*Figure 1A*). We then detected the nucleocytoplasmic distribution of the endogenous GDOWN1 in the fractionated cell lysates from various human and mouse cell lines by Western Blotting (WB) using two *KO*-verified GDOWN1 antibodies (*Figure 1—figure supplement 1A*). The results clearly indicated that the endogenous GDOWN1 was predominantly located in the cytoplasmic fractions in all the human and mouse somatic cell lines tested and only a small fraction was seen in the nuclear extract of the mouse embryonic stem cell line E14TG2a (*Figure 1B*, *Figure 1—figure supplement 1B*). These data indicate that GDOWN1 is a strictly cytoplasmic protein in various human and mouse somatic cells.

Combining GDOWN1's potential nuclear functions and our observation of its cytoplasmic localization, it is reasonable to hypothesize that GDOWN1 is a nucleocytoplasmic shuttling protein that transiently functions in the nucleus in a tightly controlled mode. Most of the nucleocytoplasmic shuttling proteins contain a nuclear export signal (NES) and a typical NES contains a hydrophobic leucine-rich motif recognized by the ubiquitous transporter chromosome maintenance protein 1, CRM1 (*la Cour et al., 2004*; *Xu et al., 2010*). To test the possibility of GDOWN1 being a CRM1 cargo, HeLa and SW620 cells were treated with leptomycin B (LMB), a potent CRM1 inhibitor for efficiently blocking CRM1-NES interaction (*Kudo et al., 1999*; *Kudo et al., 1998*).

Using two *KO*-verified GDOWN1 antibodies, our WB data unambiguously demonstrated that GDOWN1 didn't accumulate in the nucleus upon LMB treatment (*Figure 1C*, *Figure 1—figure supplement 1C*). The resistance to LMB treatment implies that either GDOWN1 does not have any NES, or this treatment alone is not sufficient to achieve the nuclear accumulation of GDOWN1.

On the other hand, we employed BiFC assays to detect the interactions between GDOWN1 and its potential nuclear binding partners in live cells. An efficient interaction between the proteins of interest drives the formation of the fluorescence complementation, achieved via the covalent interactions between the two truncated parts of a fluorescent protein (*Hu et al., 2005*). Therefore, BiFC signals are irreversible once generated, making this assay beneficial for capturing the transient protein-protein interactions. To support the specificity of BiFC assays, a collection of the negative controls was provided, including three cytoplasmic proteins, VDAC1 (a mitochondrial protein), GALNT2 (a Golgi protein) and PDIA3 (an endoplasmic reticulum protein) (*Figure 1—figure supplement 1D*). Then, a series of the transcription associated proteins were tested in HeLa cells, including a Pol II subunit (RPB5) and the RPB1-CTD binding factors (RPRD1A, RPRD1B), the Mediator components (MED1, MED26), and the transcription elongation factors (SPT4 and SPT5 in DSIF complex, NELF-E in NELF complex). The results indicated that GDOWN1 interacted to all the above factors tested except for the two Mediator components, well supporting its known characteristics as a co-purified factor of Pol II and the potential functions involved in transcriptional regulation (*Figure 1D*). However, all these BiFC

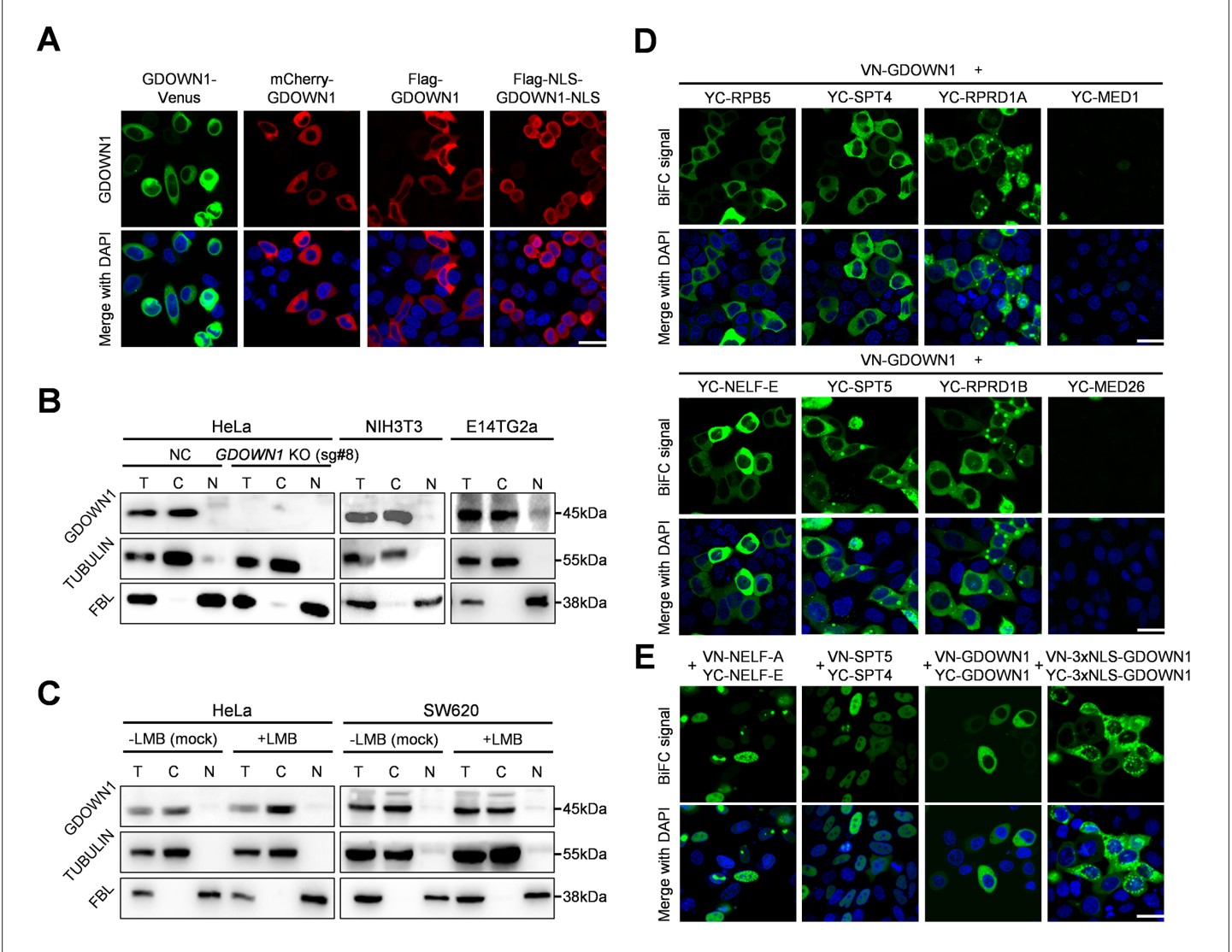

**Figure 1.** Detection of the subcellular localization of GDOWN1 or BiFC signal between GDOWN1 and some transcription related factors. (**A**) The ectopically expressed human GDOWN1 in HeLa cells was stringently localized in the cytoplasm. Human GDOWN1 proteins fused with indicated tags, including a fluorescent tag at either terminus, a Flag tag alone or together with two NLS motifs, were ectopically expressed in HeLa cells and the subcellular localization was detected by directly monitoring the fluorescent signal or by immunofluorescence assays (IF) using an anti-FLAG antibody. (**B**) The endogenous human or mouse GDOWN1 was stringently located in the cytoplasm. Each indicated cell line was fractionated to separate the cytosol from the nuclei, and the cytoplasmic fraction, C,the nuclear fraction, N, and the whole cell lysate, T (total), were further detected by WB. α-TUBULIN and FBL (a nucleolus protein) were used as markers of the cytoplasmic and nuclear fractions, respectively. (**C**) GDOWN1 remained in the cytoplasm upon LMB treatment. The indicated cell lines were subjected to either mock or LMB treatment (detailed below) before further fractionation and WB analyses. (**D**) BiFC analyses of the protein-protein interactions between GDOWN1 and its potential binding partners. (**E**) BiFC analyses of the protein-protein interactions between NEFL-E•NEFL-A, SPT4•SPT5, GDOWN1•GDOWN1 or 3xNLS-GDOWN1•3xNLS-GDOWN1. Proteins of interest were cloned and fused with either VN (the N-terminus of Venus) or YC (the C-terminus of YFP), and each of the indicated pair of plasmids was co-transfected into HeLa cells, and the confocal microscopy images were acquired 24 hr post transfection. The LMB treatment was carried out at 20 nM final concentration for 6 hr and the mock treatment was done with an equal volume of ethanol in parallel. Nuclear DNA was stained by Hoechst 33342. scale bars—30 μm. Without further labeled with details, the GDOWN1 antibody used in WB assays was the one generated from rabbits.

The online version of this article includes the following source data and figure supplement(s) for figure 1:

**Source data 1.** Raw data of WB for *Figure 1B and C*.

**Figure supplement 1.** Verification of *GDOWN1 KO* cell lines and the negative control of BiFC system.

**Figure supplement 1—source data 1.** Raw data of WB for *Figure 1—figure supplement 1A*.

**Figure supplement 1—source data 2.** Raw data of WB for *Figure 1—figure supplement 1B*.

**Figure supplement 1—source data 3.** Raw data of WB for *Figure 1—figure supplement 1C*.

signals were shown in the cytoplasm, yet in the parallel tests, the interaction signals between NELF-E•NELF-A, SPT4•SPT5 pairs were both exclusively present in the nucleus as expected (*Figure 1E*). Meanwhile, our BiFC assays detected the self-interaction of GDOWN1 in the cytoplasm, suggesting that GDOWN1 may form homodimers or oligomers in cells. It was reported that the transcription regulator RYBP contained three potent and functionally independent NLS motifs (*Tan et al., 2017*), and when attached to GDOWN1, the BiFC signal of the 3xNLS-GDOWN1 dimers mainly remained in the cytoplasm (*Figure 1E*). These results support GDOWN1's potential functions in transcriptional regulation, while the stringent cytoplasmic localization of the BiFC signals indicates that GDOWN1 is restricted from entering the nucleus under the regular cell culture conditions. Thus, our data further confirm that the nuclear entry of GDOWN1 is subjected to a tight regulation. Apparently, alleviation of this restriction will be a prerequisite for permitting GDOWN1 to execute its transcriptional regulatory functions in the nucleus.

## Human GDOWN1's cytoplasmic localization is determined by two distinct types of localization regulatory signals

Next, we constructed a series of GDOWN1 mutants to screen for localization regulatory signals by monitoring the changes in the subcellular localization by itself or together with other proteins. Based on the information of the predicted secondary structure (*Figure 2—figure supplement 1A*), GDOWN1 was truncated into three parts at its structurally flexible coiled regions (the N-terminus, namely *m1*; the middle part, *m2*; the C-terminus, *m3*), fused with a Flag-VN tag in a BiFC vector or with an intact Venus to monitor their dynamic localization in the presence or absence of LMB treatment (*Figure 2A and B*). Consistent to the above cell fractionation results, both the ectopically expressed full length GDOWN1, and the BiFC signal between the full length of GDOWN1 and NELF-E remained in the cytoplasm in the presence of LMB (*Figure 2B*, *Figure 2—figure supplement 1B*). *m1* was partially located in the nucleus (*Figure 2B*, *left panel*; *Figure 2—figure supplement 1B*), and the BiFC signal of *m1*•NELF-E was completely nuclear localized (*Figure 2B*, *right panel*). The other two GDOWN1 fragments, *m2* and *m3*, remained the cytoplasmic localization on their own or together with NELF-E, which was the same as the full-length protein. Interestingly, both *m2* alone and *m2*•NELF-E BiFC signals were translocated into the nucleus upon the LMB treatment, while either *m3* alone or the *m3*•NELF-E signal did not respond to LMB at all (*Figure 2B*, *Figure 2—figure supplement 1B*). The quantitative and statistical analyses of these confocal images confirmed that the change in the nucleocytoplasmic distribution of *m2* upon the LMB treatment was significant (*Figure 2B*, *Figure 2—figure supplement 1B*). These results clearly indicate that the middle part of GDOWN1 contains an NES motif. Given that the translocation of GDOWN1 into the nucleus may not be an autonomous and efficient process, we reasoned that monitoring the nuclear accumulation of the BiFC signals between GDOWN1 and its nuclear binding partners (such as NELF-E) would have the advantage for better mining and demonstrating the nuclear translocation potential of GDOWN1. Thus, the above BiFC system was further employed for screening the putative localization regulatory motif(s). The conserved sequence of a classical NES for CRM1 recognition was known as $\Psi$-$(x)_{1-3}$-$\Psi$-$(x)_{1-3}$-$\Psi$-$(x)_{1-3}$-$\Psi$ ($\Psi$ stands for L, I, V, M, or F, x can be any amino acid) (*la Cour et al., 2004*; *Xu et al., 2012*). We generated a '*m1 +m2*' mutant, namely *m4*, and found it responded to LMB (*Figure 2B*). We then tested a series of GDOWN1 truncation mutants on the basis of *m4* to identify the functional NES motif (*Figure 2—figure supplement 1C*, *m5-m7*), and further confirmed that a putative NES motif located between amino acids 191–201 was responsible for the LMB responsiveness. Mutation of the four hydrophobic amino acids within this region completely abolished the NES activity (*Figure 2B*, *m4 and m4\**). By co-immunoprecipitation and BiFC assays, we confirmed that GDOWN1 interacted to CRM1/RAN, the core components of the protein nuclear export machinery (*Figure 2C*). These results prove that GDOWN1 indeed contains a classical CRM1-dependent NES motif, and meanwhile suggest that the C-terminus of GDOWN1 contains a regulatory motif responsible for the observed resistance activity of the full-length GDOWN1 to LMB treatment.

When *m1* and *m3* parts were combined to generate *m8*, it was not subjected to LMB-dependent nuclear accumulation. However, when the very end of the C-terminus was chopped off, it became LMB-responsive, which led to the identification of the second NES (*Figure 3A and B*, *m8 and m9*, *Figure 3—figure supplement 1A*). Further mutant screening identified the second NES located between amino acids 332–340 (*Figure 3A and B*, *Figure 3—figure supplement 1B–D*, *m9\**, *m12*,

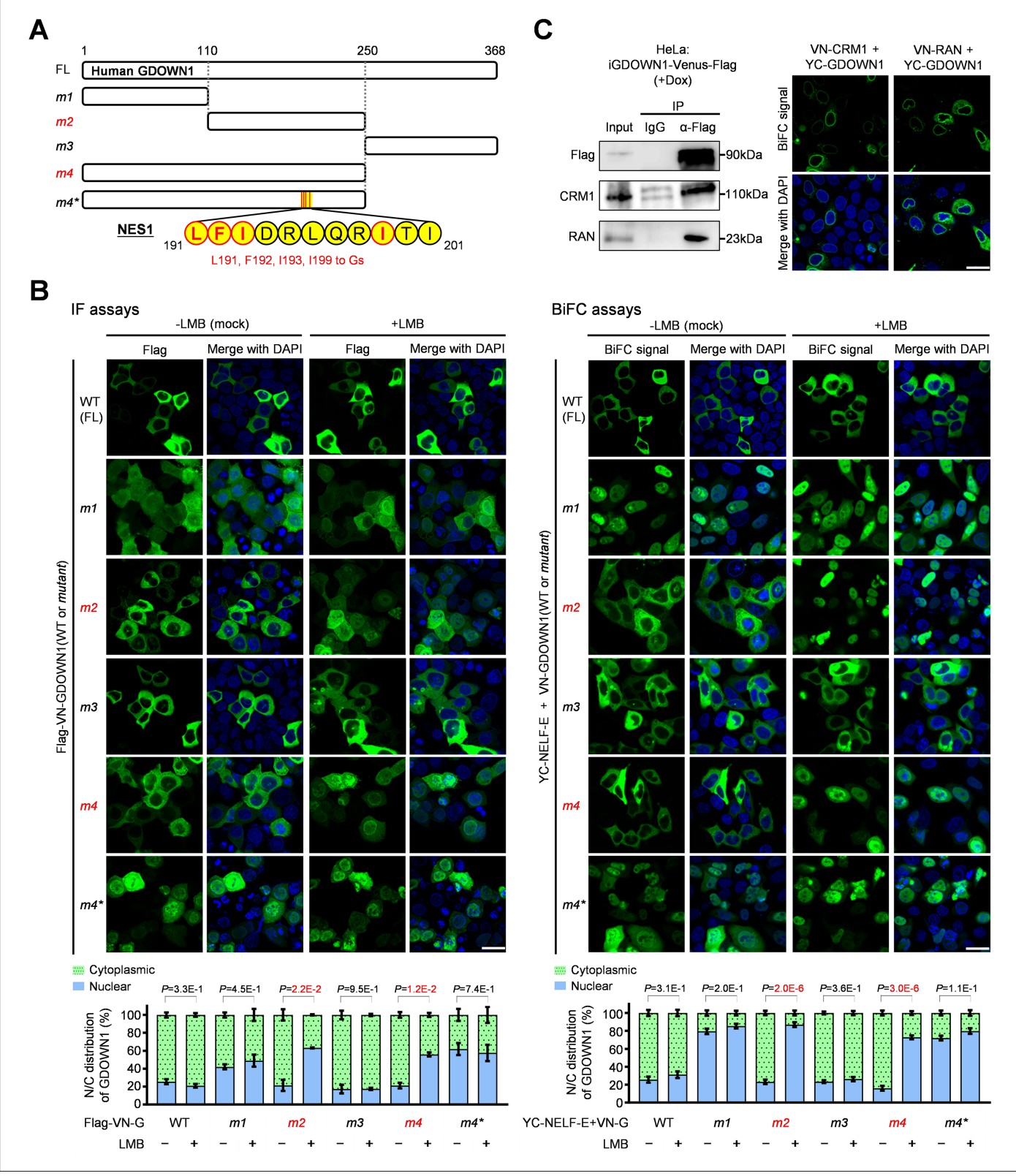

**Figure 2.** Identification of the Nuclear Export Signal (NES) motifs in GDOWN1. (**A**) A diagram of human GDOWN1 and its mutants used in the IF or BiFC-based motif screening analyses. The mutants whose names are marked in red are the ones translocated into the nucleus in response to LMB treatment. The sequences of the identified NES motifs are shown in yellow circles and the positions are labled on each side, and the core amino acids selected for mutagenesis are highlighted in red. (**B**) Identification of the NES motifs in GDOWN1 via IF or BiFC-based screening analyses. Left panel:

*Figure 2 continued on next page*

*Figure 2 continued*

HeLa cells were transiently transfected with a plasmid carrying Flag- WT or mutant GDOWN1 as indicated, and further subjected to either mock or LMB treatment, the subcellular localization was detected by IF using a Flag antibody; Right panel: HeLa cells were transiently transfected with two BiFC plasmids, YC-NELF-E and VN-WT or mutant GDOWN1 as indicated (VN—the N-terminus of Venus; YC—the C-terminus of YFP), and further subjected to either mock or LMB treatment before signal detection by a confocal microscope. The nucleocytoplasmic distribution of the fluorescent signals was quantified using ImageJ and shown at the bottom. The *P* values were calculated via a *t*-test using the built-in tools in Graphpad Prism8, n≥2, significant: *P*<0.05. (**C**) Detection of the interaction between GDOWN1 and CRM1 or RAN by IP-WB or BiFC assays. Left panel: HeLa cells stably expressed GDOWN1-Venus-Flag were employed for IP experiment using a Flag antibody or IgG and further detected by WB with the indicated antibodies; Right panel: BiFC analyses of GDOWN1•CRM1/RAN interactions. HeLa cells were transfected with YC-GDOWN1 and VN-CRM1 or RAN. The LMB treatment was carried out at a final concentration of 20 nM for 6 hr and the mock treatment was done with an equal volume of ethanol in parallel. The nuclear DNA was stained with Hoechst 33342. scale bars—30 µm.

The online version of this article includes the following source data and figure supplement(s) for figure 2:

**Source data 1.** Raw data used for the statistical analyses presented in *Figure 2B*.

**Source data 2.** Raw data of WB for *Figure 2C*.

**Figure supplement 1.** Detection of the subcellular localization of the indicated *GDOWN1* mutants.

**Figure supplement 1—source data 1.** Raw data used for the statistical analyses presented in *Figure 2—figure supplement 1B*.

**Figure supplement 1—source data 2.** Raw data used for the statistical analyses presented in *Figure 2—figure supplement 1C*.

*m13*). Taken together, we confirm that GDOWN1 is a CRM1 cargo containing two classical CRM1-responsive NES motifs.

The distinct responsiveness of the *m8* and *m9* parts of GDOWN1 to LMB treatment clearly indicated that the C-terminus of GDOWN1 contained a CRM1-independent, cytoplasmic localization signal. The key amino acids were then examined in the BiFC reporter system by screening a series of C-terminal truncation or deletion mutants (*Figure 3—figure supplement 1C, D*, *m14-m16*). It turned out that deletion of the amino acids 352–361 abolished this cytoplasmic localization regulatory activity and switched GDOWN1 into a LMB-responsive manner (*Figure 3A, B*, *m10*, *Figure 3—figure supplement 1A and B*; *Figure 1*, *m10*). After testing a series of combinations of point mutations, we found that mutations of the three arginines (R352, R354, and R357) were efficient to abolish the above cytoplasmic localization activity of GDOWN1 in the presence of LMB (*Figure 3A, B*, *m11*, *Figure 3—figure supplement 1B–D*, *m11*, *m17*, *m18*). Due to its potent cytoplasmic retention activity, we named this region (352–357 aa) Cytoplasmic Anchoring Signal, CAS.

To further elucidate the working mechanism of the CAS motif, we generated a pair of stable HeLa cell lines that inducibly expressed either the wild type GDOWN1 (WT-Venus) or its *CAS* mutant (*mCAS*-Venus) (*Figure 3—figure supplement 2A*). In these stable cell lines, the dynamic localizations of GDOWN1 were monitored and the consistent results were obtained (*Figure 3—figure supplement 2B*). Interestingly, the confocal microscopy images demonstrated that the wild type GDOWN1 accumulated around the nuclear membrane, as if these molecules attempted to burst through this last defense line into the nucleus, while the *CAS* mutant lost this 'ring-form' accumulation at the nuclear periphery, and became widely scattered all over the cytoplasm (*Figure 3C*). We hypothesized that the Venus signal enriched around the nuclear periphery might be an indicator of GDOWN1's association with the Nuclear Pore Complex (NPC). Due to the complicated composition of NPC, we detected the interaction of GDOWN1 to several representative NPC components via BiFC assays. RAE1 and NUP50 are two NPC components typically assembled within the cytoplasmic filaments and the nuclear baskets, respectively. BiFC results indicated that the wild type GDOWN1 strongly interacted to RAE1 at the cytoplasmic side of the nuclear membrane and in the cytoplasm while this interaction was drastically weakened in the *CAS* mutant (*Figure 3D*), suggesting that the CAS motif was involved in the GDOWN1•NPC interaction. More interestingly, the BiFC signal of the wild type GDOWN1 and NUP50 was very weak and randomly distributed throughout the cytoplasm, but when the CAS motif was mutated, this interaction signal was specifically translocated into the nucleus, especially at the inner face of the nuclear membrane where NUP50 naturally located (*Figure 3D*). The IP results also demonstrated that the wild type GDOWN1 interacted with the cytoplasmic NPC component, NUP214, while the *CAS* mutant lost this interaction (*Figure 3E*). Overall, these results demonstrated that GDOWN1 specifically interacted to the cytoplasmic NPC components, while the *CAS* mutant reduced this binding affinity and simultaneously enhanced the interaction of GDOWN1 to the nuclear NPC components. Due to the irreversible nature of BiFC signal, the nuclear signal of

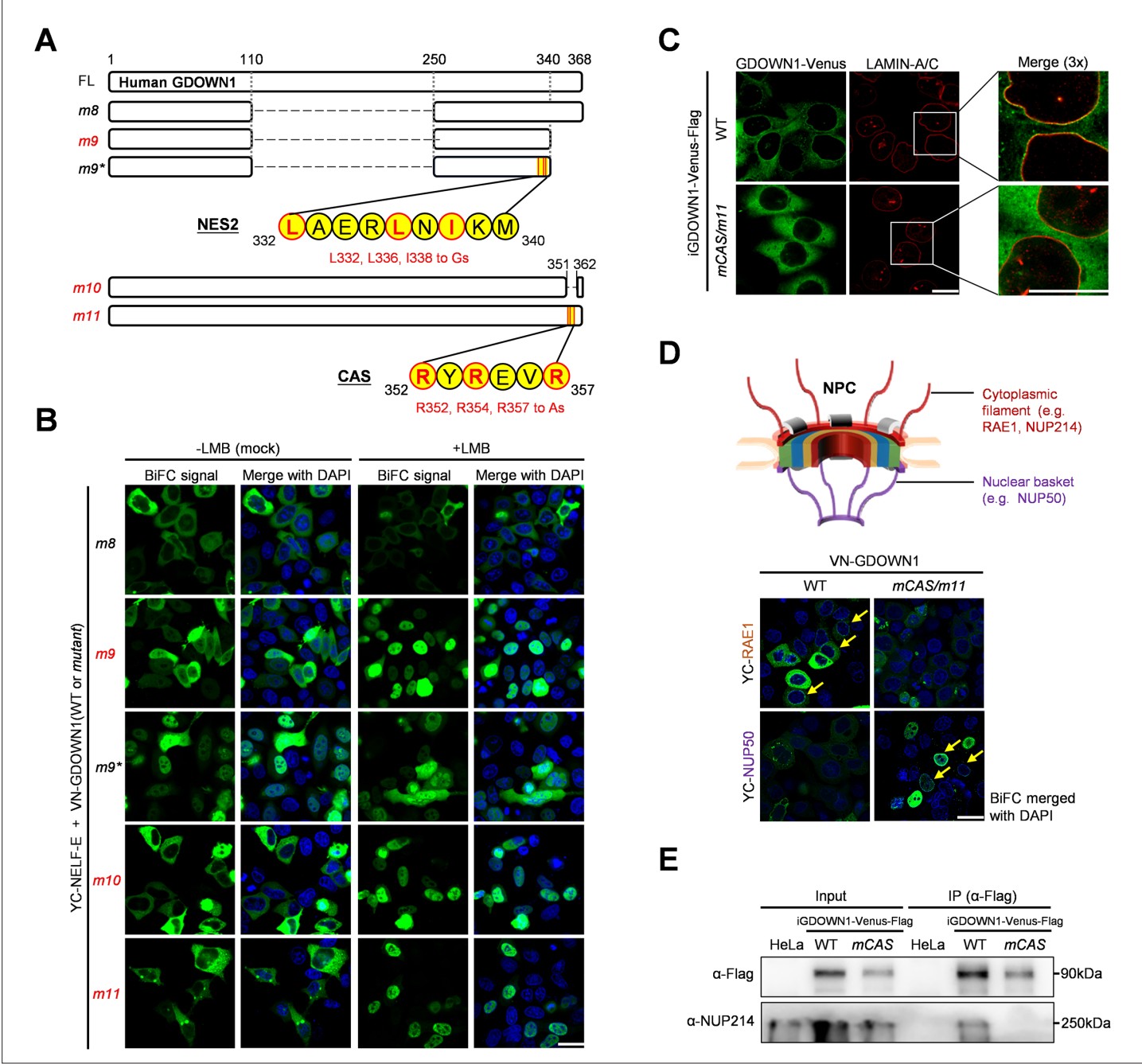

**Figure 3.** Identification and mechanistic analyses of the Cytoplasmic Anchoring Signal (CAS) motif in GDOWN1. (**A**) A diagram of human GDOWN1 and its mutants used in the BiFC-based motif screening analyses. The mutants whose names are marked in red are the ones translocated into the nucleus in response to LMB treatment. The sequences of the identified NES or CAS motif are shown in yellow circles and the core amino acids selected for mutagenesis are highlighted in red. (**B**) Identification of the second NES and the CAS motif in GDOWN1 via BiFC-based screening analyses. The experiments were carried out in the same way as described in *Figure 2B*. (**C**) The enrichment of GDOWN1 at the nuclear pore region was regulated by the CAS motif. HeLa cells stably expressing the wild type GDOWN1 (WT-Venus) or the *CAS* mutant (*mCAS*-Venus) were used for detection. The nuclear membrane was approximately represented via IF using an antibody against the nuclear lamina (α-LAMIN-A/C). Confocal Images were collected and further zoomed in for 3 folds to show more details of the nuclear membranes. (**D**) BiFC analyses of the interactions between GDOWN1 and some subunits of NPC in HeLa cells. Upper panel: a simplified diagram of an NPC; lower panel: BiFC results between GDOWN1 (or its *CAS* mutant) and the indicated NPC components. (**E**) Detection of the interaction between GDOWN1 and NUP214 by IP-WB. Parental Hela cells or HeLa cells stably expressed GDOWN1(WT or *mCAS*)-Venus-Flag were employed in IP experiment using a Flag antibody and further detected by WB with indicated antibodies. The LMB treatment was carried out as previously described. The nuclear DNA was stained with Hoechst 33342. The scale bars represent 30 μm except for the ones in **C** represent 15 μm.

*Figure 3 continued on next page*

*Figure 3 continued*

The online version of this article includes the following source data and figure supplement(s) for figure 3:

**Source data 1.** Raw data of WB for *Figure 3E*.

**Figure supplement 1.** Detection of the subcellular localization of the indicated *GDOWN1* mutants.

**Figure supplement 1—source data 1.** Raw data used for the statistical analyses presented in *Figure 3—figure supplement 1A*.

**Figure supplement 1—source data 2.** Raw data used for the statistical analyses presented in *Figure 3—figure supplement 1B*.

**Figure supplement 1—source data 3.** Raw data used for the statistical analyses presented in *Figure 3—figure supplement 1D*.

**Figure supplement 2.** Detection of the expression and the subcellular localization of the *CAS* mutant.

**Figure supplement 2—source data 1.** Raw data of WB for *Figure 3—figure supplement 2A*.

**Figure supplement 2—source data 2.** Raw data used for the statistical analyses presented in *Figure 3—figure supplement 2B*.

*mCAS*-GDOWN1•NUP50 interaction was a clear indication of a successful capture of this GDOWN1 mutant in the nucleus, while its wild type counterpart was restricted in the cytoplasm. The above data highlight the crucial role of the CAS motif on locking GDOWN1 in the cytoplasm, presumably through anchoring GDOWN1 to the cytoplasmic components of NPC, and imply that any cellular strategy of preventing CAS function will potentially switch GDOWN1 from a stringent cytoplasmic localized protein into a nucleocytoplasmic-shuttling protein.

## The NES and CAS motifs in Gdown1 are functionally interconnected and both conserved during evolution

Based on the structural prediction of GDOWN1, its CAS motif is located within the disordered region near the carboxyl-terminus, which makes it difficult to obtain reliable structural information to predict the potential CAS-NES interaction (*Figure 4—figure supplement 1A*). Indeed, a previous report, carrying out chemical crosslinking with mass spectrometry readout (CX-MS) to analyze Gdown1-Pol II interaction, did not provide any information about the CAS region (*Jishage et al., 2018*). To clarify the functional relationship between the CAS and NES motifs, we transiently expressed GDOWN1-Venus or its localization motif mutants that carried combinations of the mutated key amino acids identified above, and tested their subcellular localization and LMB responsiveness (*Figure 4A*). When both NES2 and CAS were mutated to allow NES1 alone to function, the mutant *GDOWN1* performed as a typical CRM1-cargo, and on the other hand, the *NES1* mutant maintained the same cytoplasmic localization and LMB resistance activity as the wild type GDOWN1 (*Figure 4A*, a-c). Thus, NES1 was a functional NES motif working independently to NES2. Similarly, we proved that NES2 was a functional NES as well, although somewhat weaker than NES1 (*Figure 4A*, d and b). The subcellular distribution of GDOWN1 in these images was quantified and plotted in *Figure 4B*. It turned out NES1 better responded to CRM1 than NES2. The *NES2* mutant regularly remained in the cytoplasm while it did not resist to LMB treatment as well as the wild type, suggesting that the cytoplasmic localization activity of CAS might be partially interfered (*Figure 4A*, e). Double mutations in both NES motifs made GDOWN1 distributed in both the cytoplasm and the nucleus, and did not respond further to LMB, which proved that the entire GDOWN1 contained two NES motifs, and again mutations in NES2 partially abolished the CAS activity (*Figure 4A*, f). Compared with the wild type, the *CAS* mutant responded well to the LMB treatment (*Figure 4A*, g). Further mutating the CAS motif on top of NES1 or NES2 increased the portion of the nuclear GDOWN1 upon LMB treatment (*Figure 4A*, d and c; b and e), suggesting that CAS is the main motif to keep GDOWN1 in the cytoplasm via a CRM1-independent manner. Consistently, in the stable cell lines, we found that the *NES2* mutant lost the perinuclear staining, which was very similar to the phenotype of the *CAS* mutant (*Figure 4C*). The above data from the intrinsic motif analyses demonstrate that each one of the two NES motifs of GDOWN1 acts as an independent CRM1-regulated motif and the function of CAS motif partially depends on the existence of NES2. Taken together, GDOWN1 is identified as a nucleocytoplasmic shuttling protein subjected to both CRM1-dependent and CRM-independent regulation and the two layers of regulation are functionally coupled.

Since the nucleocytoplasmic shuttling effect of Gdown1 was reported in *drosophila*, we evaluated the conservation of its localization regulatory mechanisms across species. The Clustal Omega analyses were carried out to compare the Gdown1 sequences from various representative species (fly, zebrafish,

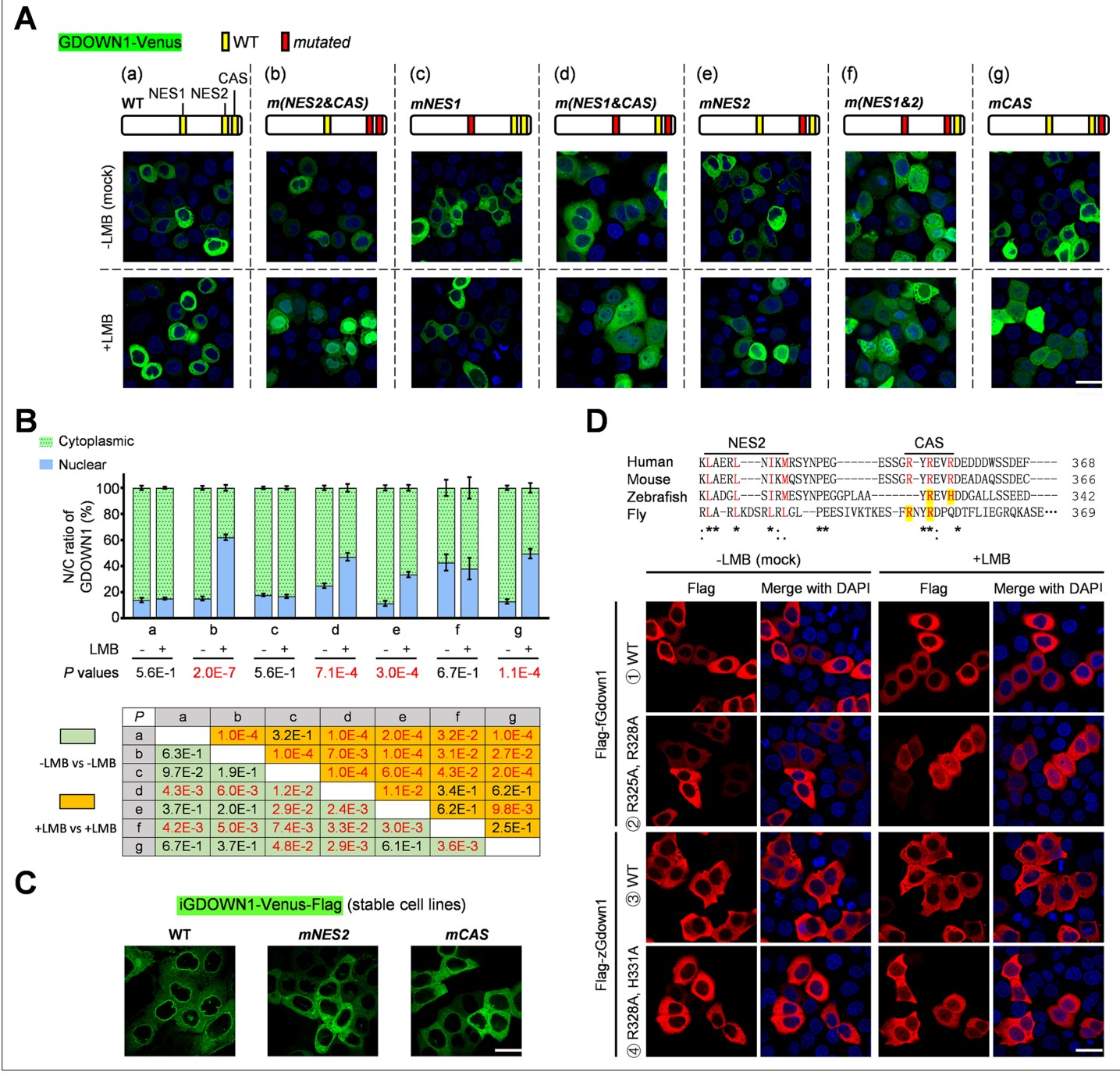

**Figure 4.** The working mechanisms and conservation of the binary localization regulatory apparatus in Gdown1. (**A**) Dissection of the functional independence and interplay among CAS and NES motifs. The wild type GDOWN1 or the indicated *CAS* or *NES* mutants carrying point mutations were fused with Venus and ectopically expressed in HeLa cells. The cells were subjected to mock or LMB treatment the same as described in ***Figure 1***. The schematic diagram of each mutant is shown on the top side of the corresponding representative confocal microscopy images. (**B**) The quantitative and statistical analyses of A. The nucleocytoplasmic distribution of the fluorescent signals for all the mutants shown in A was quantified using ImageJ and shown on the top panel. For the statistical analyses, the P values about the distribution changes in response to LMB for each mutant and about the differences between each pair of the samples were calculated via t-test using the built-in tools in Graphpad Prism8, n=4. The P values that are smaller than 0.05 (significant) were highlighted in red. (**C**) Confocal images demonstrating the subcellular localization of GDOWN1 in the indicated stable cell lines upon Dox induction for 2 days. (**D**) The function of the NES and CAS motifs was very conservative from zebrafish and *drosophila* to mammals. Upper panel: the sequence alignment of the putative NES2-CAS regions of Gdown1 proteins from the indicated species (*Homo sapiens*, NP_056347.1, *Mus musculus*, NP_848717.1, *Danio rerio*, NP_001333109.1, *Drosophila melanogaster*, NP_650794.1). '*'—identical in all species analyzed; ':'—highly conserved; '.'—moderately conserved. Lower panel: the dynamic subcellular localization of the wild type or *CAS* mutants of zebrafish (zGdown1) and

*Figure 4 continued on next page*

*Figure 4 continued*

fly (fGdown1) was detected by IF experiments. The plasmids expressing the indicated proteins were transfected into HeLa cells and the LMB treatment was carried out as previously described. The nucleocytoplasmic distribution of the fluorescent signals was quantified using ImageJ and shown on the bottom. scale bars—30 µm.

The online version of this article includes the following source data and figure supplement(s) for figure 4:

**Source data 1.** Raw data used for the statistical analyses presented in *Figure 4B*.

**Figure supplement 1.** The structural prediction of GDOWN1 and the conservation analyses of Gdown1 across species.

**Figure supplement 1—source data 1.** Raw data used for the statistical analyses presented in *Figure 4—figure supplement 1C*.

**Figure supplement 1—source data 2.** Raw data used for the statistical analyses presented in *Figure 4—figure supplement 1D*.

mouse, and human). The NES motifs are modestly conserved across these species with the key hydrophobic amino acids roughly present in fly and zebrafish Gdown1 proteins (*Figure 4D*, top, *Figure 4—figure supplement 1B*). In terms of the CAS motifs, there is no difference between mouse and human, while there is only one or two key arginines remained present in the putative CAS motifs of zebrafish and fly Gdown1 proteins, respectively. When ectopically expressed in HeLa cells, fly and zebrafish Gdown1 proteins also located stringently in the cytoplasm and resisted to LMB treatment as same as their human counterpart (*Figure 4D*, *Figure 4—figure supplement 1C*). When the conserved amino acids in the putative CAS motifs of fly and zebrafish Gdown1 proteins were mutated, these mutants became partially nucleus localized upon LMB treatment (*Figure 4D*, *Figure 4—figure supplement 1C*), indicating that fly and zebrafish Gdown1 also contained functional NES and CAS motifs. In addition, the results from BiFC analyses demonstrated that fly and zebrafish Gdown1 proteins were able to interact to human NELF-E in the cytoplasm, indicating that these orthologs in lower animals were structurally conservative to human GDOWN1 (*Figure 4—figure supplement 1D*). Different from the mammalian counterpart, the BiFC signals between fly or zebrafish Gdown1 and NELF-E were partially translocated into nucleus in the presence of LMB, and when CAS regions were mutated, these BiFC signals were completely present in the nucleus, indicating that the regulatory effect of CAS in fly and zebrafish Gdown1 was present but not as potent as that in human (*Figure 4—figure supplement 1D*). Furthermore, we found that *Drosophila CAS* mutant appears to accumulate more in the nucleus than the mutated zebrafish protein in the presence of LMB (*Figure 4D*, *Figure 4—figure supplement 1C and D*). We surmised that the phenomenon is because the CAS motif in fly Gdown1 is more conserved than which in zebrafish. The above results demonstrate that both the CRM1-dependent and CRM1-independent regulatory mechanisms of Gdown1 are well conserved across the various species from flies to human, while during evolution, the cytoplasmic anchoring effect of the CAS motif seems to have been enhanced to strengthen the regulation of Gdown1's subcellular localization.

## Nuclear-accumulated GDOWN1 reduces the total Pol II and the global transcription levels, and prevents cell growth

The great effort devoted by the cells to prevent Gdown1 from entering the nucleus strongly implies that it is essential to stringently control the nuclear activities of Gdown1. To help explore the outcome of Gdown1's nuclear accumulation in somatic cells, we generated a nuclear localized, full-length human *GDOWN1* mutant by mutating all the ten key amino acids identified in the three motifs of NES and CAS (highlighted in red in *Figures 2A and 3A*, simply named the *10M* mutant). The wild type GDOWN1 or the GDOWN1(*10M*) was fused with Venus, and further with or without an NLS, in a commercial pTripZ vector to achieve the doxycycline (Dox)-inducible expression. The four corresponding stable HeLa cell lines were generated. The main experimental procedures are demonstrated in a diagram of *Figure 5A*. The confocal images after a one-day induction were shown and the *10M* mutant was evenly distributed in cells and further addition of an NLS motif switched GDOWN1 into a complete nuclear localized protein (NLS-*10M*) (*Figure 5B*, i). These stable cell lines were generated by collecting the pool of cells survived from the puromycin selection, which turned out to be heterogenous that both Venus⁺ and Venus⁻ cells were present upon Dox induction. The benefit of using such heterogenous cell pools instead of the single clones hereby was that the co-cultured Venus⁻ cells (mainly expressing the very low levels of GDOWN1-Venus) could serve as the internal negative controls for a parallel comparison. When Dox was continuously supplemented in the culture medium, the fluorescence intensity in the Venus⁺ cells and their ratio to the whole population reached nearly maximum

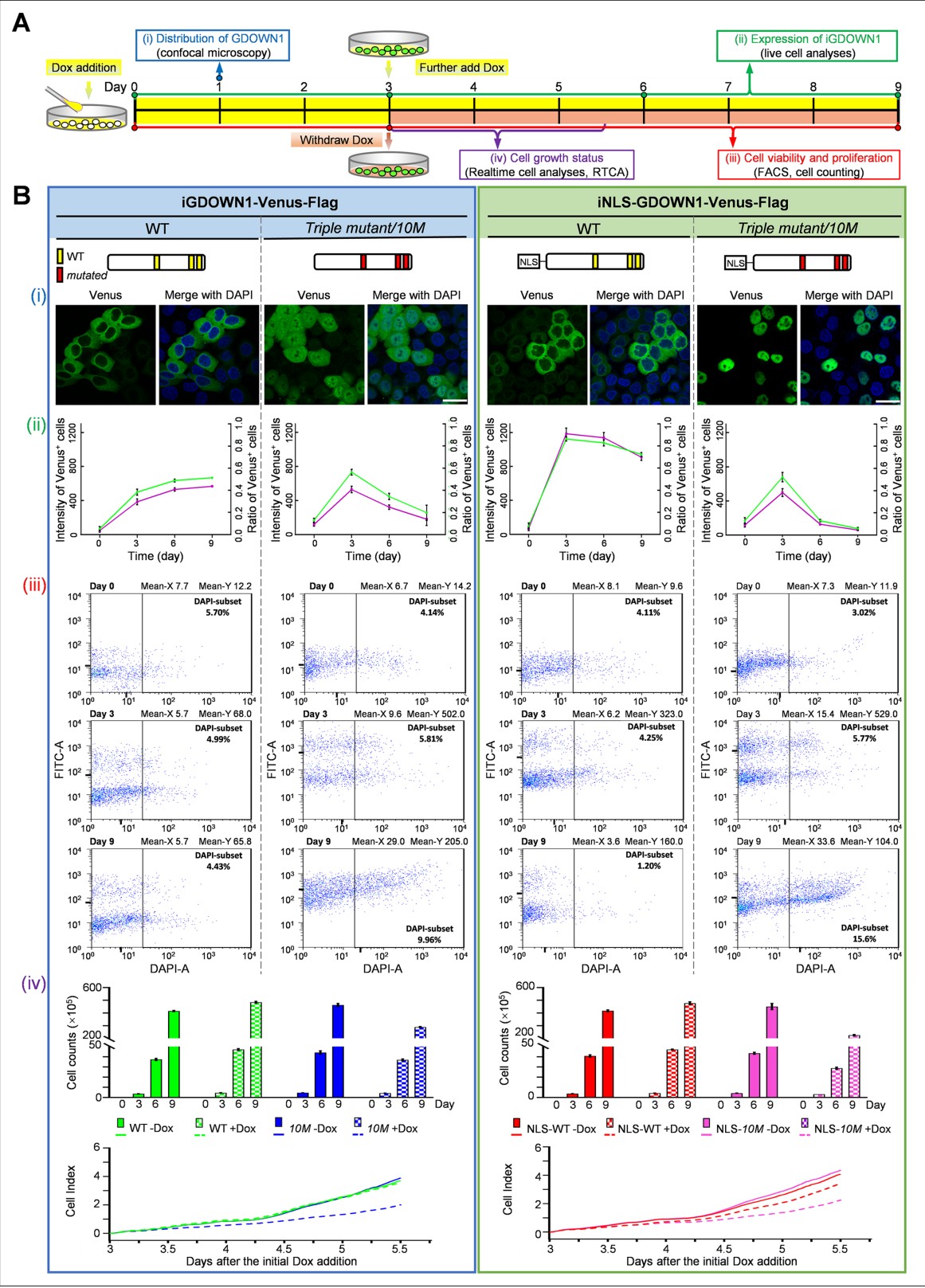

**Figure 5.** Massive accumulation of GDOWN1 in the nucleus slows down cell growth and may trigger cell death. HeLa cells stably and inducibly expressing GDOWN1- or NLS[sv40]-GDOWN1-Venus-Flag, either wild type or the *10M* mutants were used for detection. 'i' stands for inducible and Dox was used as the inducer. (**A**) The experimental scheme of the comprehensive analyses of the GDOWN1 expressing cell lines (**B**) The schematic diagrams of the GDOWN1 variants are shown on the top panel. (i) Confocal images are presented to show the subcellular localization of the indicated cells upon

*Figure 5 continued on next page*

*Figure 5 continued*

Dox induction for 1 day. The nuclear DNA was stained with Hoechst 33342. scale bars—30 µm (ii) The changes of the fluorescence intensity and the ratio of Venus$^+$ cells were monitored upon the induction of GDOWN1 or its mutants. Images were acquired by Cytation 5 and data were further analyzed by Gen5. (iii) The cell death and the changes in the fluorescence intensity were detected via flow cytometry. Cells were induced by Dox for 3 days to reach the maximum expression and continuously cultured for 6 days in the absence of Dox. Then, the cells were subjected to a quick DAPI staining, followed by the flow cytometry analyses. The mean values of the FITC signal (indicating the expression levels of GDOWN1-Venus proteins) and of the DAPI signals were labeled on each graph. Meanwhile, cells were counted on days 0, 3, 6, and 9, and the growth curves were plotted and shown at the bottom. (iv) Two methods were employed to generate the cell growth curves, including the direct cell counting by a cell counter at the indicated time points (the top panels, n≥3) or detection of the cell growth status by a live cell analyzer (the bottom panels). The same amounts of the cells from each cell line were re-plated in an E-plate 16 (Agilent) after a 3 day Dox induction, and subjected to the RTCA analyses in the presence or absence of Dox (0.25 µg/mL) for another 2.5 days. The real time cell index parameters were recorded and plotted by RTCA.

The online version of this article includes the following source data and figure supplement(s) for figure 5:

**Source data 1.** Raw data used for the line chart presented in *Figure 5B* (ii).

**Source data 2.** Raw data of the cell counting presented in *Figure 5B* (iv).

**Source data 3.** Raw data used for the line chart presented in *Figure 5B* (iv).

**Figure supplement 1.** The expression analyses by WB and the cell index measurement by RTCA for the indicated GDOWN1 expressing cell lines.

**Figure supplement 1—source data 1.** Raw data of WB for *Figure 5—figure supplement 1A*.

around day 3, and remained stable hereafter in the cell lines expressing the wild type GDOWN1 (*Figure 5B*, ii). However, these values were significantly reduced in the cell lines expressing the nuclear localized GDOWN1(*10M*) mutants, especially in the NLS-GDOWN1(*10M*)-Venus cells which almost completely lost the Venus signal on day 9 after the initial Dox addition, suggesting that the accumulation of GDOWN1 in the nucleus was unfavorable for the cell growth (*Figure 5B*, ii). To dissect the underlined reasons for the signal loss, we comprehensively evaluated the growth status of the cells upon expressing either the cytoplasmic or the nuclear localized GDOWN1. The cells on day 0 (Dox addition), day 3 (Dox withdrawal), and day 9 were analyzed by FACS. The results demonstrated that the cells expressing the cytoplasmic GDOWN1 only showed the basal levels of cell death (indicated by the DAPI$^+$ subgroup), while the cells expressing the nuclear GDOWN1(*10M*) showed a drastic reduction of expression (indicated by the decreased FITC values), and simultaneously those Venus$^+$ cells mainly contributed to the significant increased death rate at the later time point (*Figure 5B*, iii). The expression levels of the wild type GDOWN1 and its mutants in these cell lines were tested by WB, and the expression level of the NLS-GDOWN1(*10M*)-Venus was comparable to that of the endogenous GDOWN1 (*Figure 5—figure supplement 1A*). In addition, the results from the cell counting and the real-time cell analysis assays (RTCA) also demonstrated that the cells expressing the *10M* mutant had severe defects in their growth and proliferation (*Figure 5B*, iv, *Figure 5—figure supplement 1B*). The above data indicate that the translocation of GDOWN1 into the nucleus inhibits cell growth, and its continuous and massive accumulation eventually causes cell death.

It was known from the *in vitro* transcription assays that Gdown1 negatively regulated Pol II transcription via competing TFIIF from binding to Pol II (*Cheng et al., 2012*; *Jishage et al., 2012*). Therefore, we reasoned the cell death effects seen here might be resulted from the GDOWN1-mediated transcriptional defects. Next, EU incorporation assays were carried out in the above four cell lines expressing WT or the *10M* mutants of GDOWN1 to monitor the live transcription. Based on the literature report, the EU signals captured following a 20 minutes pulse labeling were correlated to the overall transcription, with the majority of the signals (~80%) contributed by the Pol II transcription (*Jackson et al., 2000*). The EU signals were pseudo-colorized based on the acquired intensity. It turned out that the expression of the wild type cytoplasmic GDOWN1 did not cause any obvious change of transcription while in the cell lines expressing the nuclear GDOWN1, the EU incorporation in the Venus$^+$ cells was significantly decreased compared with the Venus$^-$ cells, indicating that GDOWN1's abundance in the nucleus was negatively correlated with the level of global transcription (*Figure 6A* and *Figure 6—figure supplement 1A*).

Next, we performed the cell fractionation and WB assays to check the possible changes of Pol II, and the total level of Pol II was detected using an antibody specifically recognizing the N-terminus of RPB1. Since the global transcription was dramatically repressed upon the massive accumulation of GDOWN1, we reasoned that the remained proteins were determined by their half-lives. Histone proteins were experimentally proven to be very stable (*Savas et al., 2012*; *Toyama et al., 2013*), so

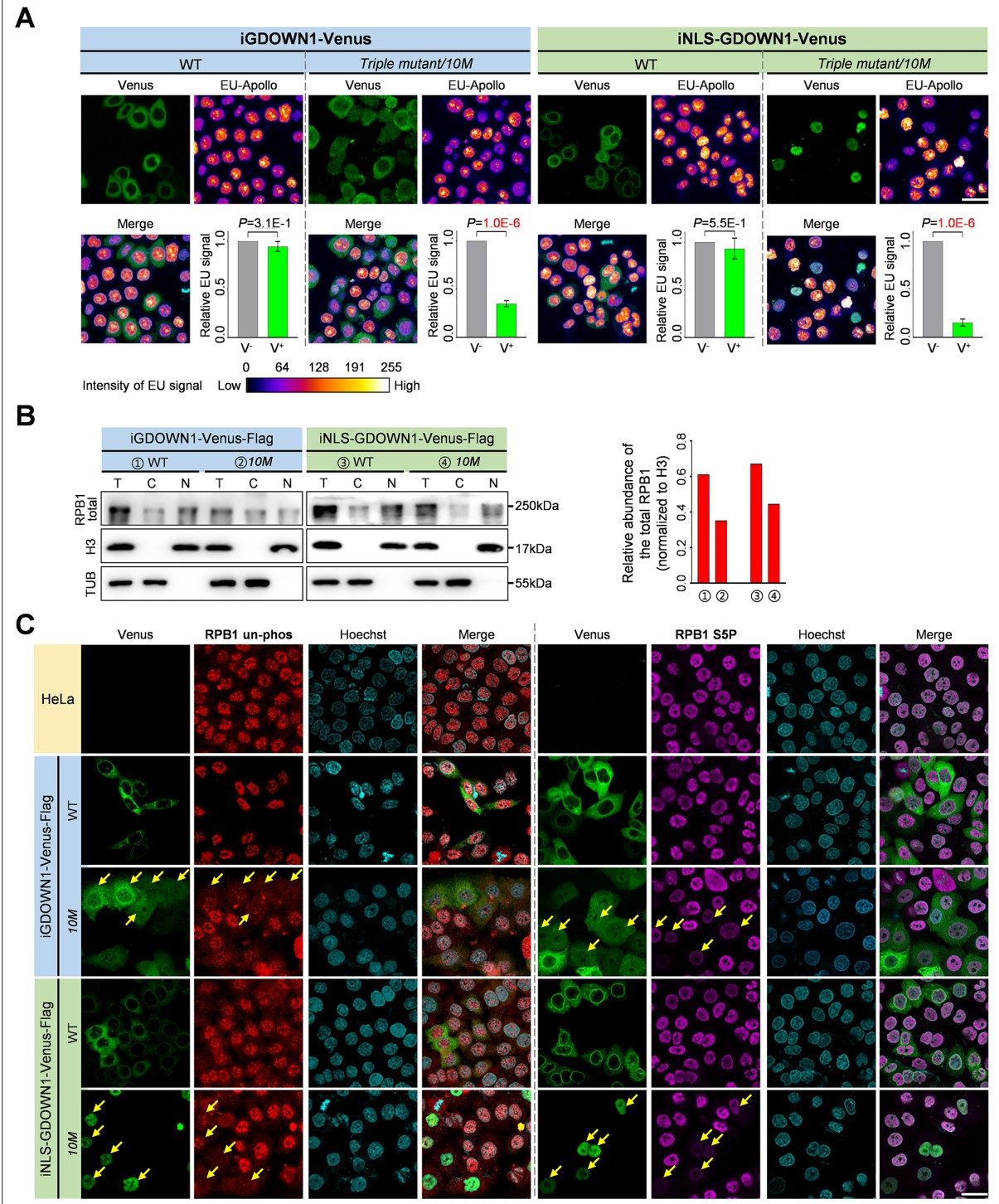

**Figure 6.** The Nuclear GDOWN1 represses the global transcription.

All the experiments shown in this figure were carried out after four days of Dox induction. (**A**) The massive accumulation of GDOWN1 in the nucleus caused the global transcriptional repression detected by the EU labeling assays. Confocal images were acquired, and the EU-Apollo signals were color-coded by ImageJ as indicated by the calibration bar shown at the bottom, based on the obtained signal intensity (the original images are shown in **Figure 6—figure supplement 1**). The averaged EU signal per cell of the Venus⁺ cells (green, V⁺) or of the Venus⁻ cells (gray, V) was shown in the graph at the lower right corner for each of the indicated cell line. The P values were calculated via a t-test using the built-in tools in Graphpad Prism8. n=4. (**B**) WB analyses of the total Pol II in the GDOWN1 expressing cell lines. The cell fractionation and the following WB analyses were carried out in the same way as previously described in **Figure 1B**. Differently, the signals of Histone H3 were served as a nuclear protein control and also for data normalization. The RPB1 level in the whole cell lysate relative to that of H3 were calculated and shown on the right. (**C**) The nuclear GDOWN1 reduces

*Figure 6 continued on next page*

*Figure 6 continued*

the levels of both the un-phosphorylated and the S5P forms of Pol II. IF experiments were carried out to detect the changes in the indicated RPB1 levels in the four indicated cell lines. Confocal images were acquired and some representative Venus[+] cells were pointed out with yellow arrows. The nuclear DNA was stained with Hoechst 33342. scale bars—30 μm.

The online version of this article includes the following source data and figure supplement(s) for figure 6:

**Source data 1.** Raw data used for the statistical analyses presented in *Figure 6A*.

**Source data 2.** Raw data of WB for *Figure 6B*.

**Source data 3.** Raw data used for the statistical analyses presented in *Figure 6B*.

**Figure supplement 1.** The extended data for *Figure 6* and the IF data quantification of the changes of various Pol II forms in response to the expression of the indicated GDOWN1.

**Figure supplement 1—source data 1.** Raw data of WB for *Figure 6—figure supplement 1B*.

**Figure supplement 1—source data 2.** Raw data used for the statistical analyses presented in *Figure 6—figure supplement 1B*.

**Figure supplement 1—source data 3.** Raw data used for the statistical analyses presented in *Figure 6—figure supplement 1D*.

we employed histone H3 as the internal control for data normalization. The results indicated that the total Pol II was clearly reduced upon the nuclear accumulation of GDOWN1 (*Figure 6B*). The 8WG16 antibody was previously confirmed to preferentially recognize the unphosphorylated RBP1 via the *in vitro* kinase assays (*Cheng and Price, 2007*). Thus, using this antibody, we found that the levels of the unphosphorylated Pol II were significantly reduced as well (*Figure 6—figure supplement 1B*). The LMB-treated cells were used as a positive control in which the cytoplasmic RPB1 was known to be increased upon the treatment (*Forget et al., 2010*; *Forget et al., 2013*). In both cases, the nucleocytoplasmic ratio of the total and the unphosphorylated forms of Pol II seemed to be unaffected.

To further dissect the detailed status of the affected Pol II, the IF assays were carried out using several Pol II antibodies with distinct specificities. Again, the 8WG16 antibody was applied to monitor the signals of the unphosphorylated Pol II, and the antibodies specifically recognizing the CTD-phosphorylated form at either the Ser5 positions (S5P) or the Ser2 positions (S2P) were employed to detect the Pol II subgroup engaged in either transcriptional initiation or productive elongation, respectively. Compared with the parental HeLa cells or the stable cell lines expressing the wild type GDOWN1, the signals of the unphosphorylated and the phosphorylated forms of Pol II were all dramatically reduced in the cell lines expressing the nuclear localized GDOWN1 (*Figure 6C*, *Figure 6—figure supplement 1C*). The image quantification analyses indicated that the S5P signals were decreased more severely than those of the S2P, suggesting that the nuclear GDOWN1 may inhibit transcription initiation more effectively than transcription elongation (*Figure 6—figure supplement 1D*). Taken together, these data demonstrate that the massive accumulation of the nuclear GDOWN1 results in the significant loss of Pol II and the global transcriptional repression.

## GDOWN1 shuttles into the nucleus in response to certain stresses and helps strengthen cellular adaptability

Next, we tested various types of reagents to search for any potential exocellular stimuli capable of triggering the nuclear translocation of the endogenous GDOWN1. No obvious change of GDOWN1's subcellular localization was observed when cells were treated with the transcriptional inhibitors DRB or Madrasin, the translational inhibitor CHX, or the inhibitors for DNA topoisomerases such as CPT or Doxorubicin (*Figure 7—figure supplement 1A*). Interestingly, we found the treatment of sodium arsenite ($NaAsO_2$) reproducibly caused the nuclear translocation of GDOWN1. $NaAsO_2$-induced nuclear translocation of GDOWN1 occurred in a dose-dependent manner and reversed upon the drug removal (*Figure 7A*). The exposure to inorganic arsenite was known to induce a global transcriptional repression (*Nelson et al., 2009*; *Rea et al., 2003*), and eventually to accumulate severe cellular toxicity, resulting in growth inhibition, DNA damage, reactive oxygen species (ROS) production, apoptosis, and/or autophagy (*Tam et al., 2020*). When cells were treated with 0.5 mM $NaAsO_2$ for 30 minutes (the prevalently used condition in the literature), nearly all cells generated stress granules (SGs) no matter GDOWN1 was competent or knocked out, indicated by the IF signals of G3BP1, a typical SG marker (*Figure 7—figure supplement 1B*). However, under a milder condition (0.1 mM of $NaAsO_2$, for 6 hr), the *GDOWN1 KO* cells also generated a great number of SGs, while SGs were only

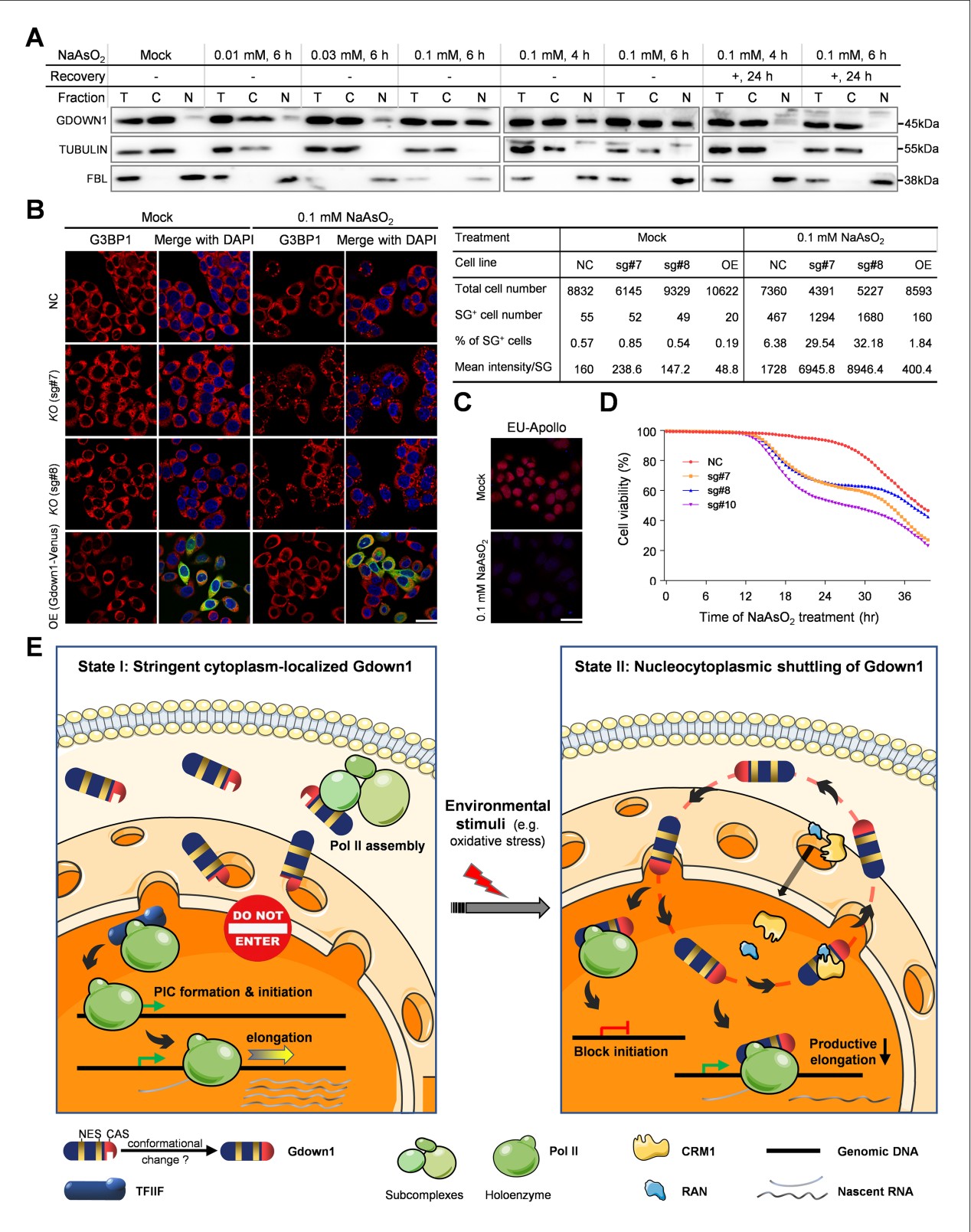

**Figure 7.** The expression levels of GDOWN1 correlate to the cellular sensitivity to NaAsO$_2$ treatment. (**A**) Upon NaAsO$_2$ treatment, a portion of the cellular GDOWN1 was subjected to a reversible translocation into the nucleus. HeLa cells were mock treated or treated with NaAsO$_2$ as indicated. In some samples, the cell culture medium was refreshed after the treatment to remove NaAsO$_2$, and the cells were further cultured for another 24 hr before harvest. The cell fractionation and the following WB analyses were carried in the same way as previously described in **Figure 1B**. (**B**) GDOWN1

*Figure 7 continued on next page*

*Figure 7 continued*

affected the formation of SGs after NaAsO$_2$ treatment. HeLa cells with *GDOWN1 KO* (sg#7, sg#8) or the negative control (sg#NC), and the cells stably and inducibly expressing iGDOWN1-Venus-Flag (OE) were employed. Each of the indicated cell lines was subjected with NaAsO$_2$ treatment at 0.1 mM for 6 hr, and the SGs were detected by IF assays using an antibody against G3BP1. The nuclear DNA was stained by Hoechst 33342. scale bars—30 µm. Left: the representative confocal images; Right: the collection of the SG parameters measured and calculated by Gen5, based on the images acquired by Cytation 5. (**C**) The total transcription level in HeLa cells was repressed upon NaAsO$_2$ treatment. HeLa cells being treated with 0.1 mM NaAsO$_2$ or mock treated were subjected to the EU-Apollo labeling assays. (**D**) Knockout of GDOWN1 made the cells more sensitive to NaAsO$_2$ stimulation. The relative cell viability of the indicated cell lines in the presence of 0.1 mM NaAsO$_2$ was monitored and calculated by Cytation 5. (**E**) A model summarizing the working and regulatory mechanisms in GDOWN1 (described in the main text).

The online version of this article includes the following source data and figure supplement(s) for figure 7:

**Source data 1.** Raw data of WB for *Figure 7A*.

**Source data 2.** Raw data used for the table presented in *Figure 7B*.

**Source data 3.** Raw data used for the line chart presented in *Figure 7D*.

**Figure supplement 1.** The subcellular localization of GDOWN1 upon various drug treatments and the formation of SGs upon the NaAsO$_2$ treatment at 0.5 mM for half an hour.

**Figure supplement 1—source data 1.** Raw data of WB for *Figure 7—figure supplement 1A*.

detected in a very small fraction in the control cells or the cells ectopically expressing the exogenous GDOWN1-Venus (*Figure 7B*). The EU staining results indicated that even under this milder NaAsO$_2$ treatment condition, the global transcription was already significantly downregulated (*Figure 7C*). Furthermore, the viability of the *GDOWN1 KO* cells was significantly less than the GDOWN1 competent counterparts (*Figure 7D*), indicating that loss of GDOWN1 made the cells hypersensitive to the cell toxicity induced by the low dose of NaAsO$_2$ treatment. Taken together, our data demonstrate that the nucleocytoplasmic localization of the native GDOWN1 is switchable in response to NaAsO$_2$-induced cellular stress and potentially to other types of unidentified cellular stimuli. Our data strongly suggest that GDOWN1-mediated transcriptional control contributes to the cellular sensitivity and adaptation to those stresses.

## Discussion

The appropriate subcellular localization of a protein determines its potential accessibility for certain cellular processes, therefore, serves as the fundamental premise for executing functions. This study is mainly focused on the exploration of human GDOWN1's subcellular localization and the associated functional and regulatory mechanisms in the somatic cells. Our results confirmed the cytoplasmic localization of Gdown1 in the cultured mammalian somatic cell lines. To demonstrate the nucleocytoplasmic shuttling properties of GDOWN1, we treated HeLa and other types of cells with a specific inhibitor of the nuclear exportin protein CRM1, LMB, with the expectation to observe its nuclear accumulation upon the treatment. Strikingly, it turned out that for all the cell lines tested, GDOWN1 remained its cytoplasmic localization in the presence of LMB, confirmed by both biochemical fractionation and the cell imaging-based assays. Furthermore, the artificial addition of NLS motifs to GDOWN1 did not efficiently promote its nuclear translocation either. Thus, we conclude that under the conventional cell culture conditions, GDOWN1 is strictly locked in the cytoplasm rather than dynamically shuttling between the cytoplasm and the nucleus (*Figure 7D*), which makes GDOWN1 remarkably different from the typical nucleocytoplasmic shuttling proteins.

Our systematic dissection of the intrinsic localization regulatory motif(s) in GDOWN1 via the detailed mutant analyses let us identify a binary localization regulatory system composed of the functionally coupled NES and CAS motifs. This delicate orchestration between CAS and NES controls the nucleocytoplasmic distribution of GDOWN1, guaranteeing the appropriate input of GDOWN1 in transcriptional regulation. The facts that both NES and CAS motifs are conservative and the CAS activity seems to be strengthened from lower to higher animals further highlight the essential role of this regulatory apparatus/mechanism in controlling Gdown1's subcellular localization and functions.

In terms of the working mechanisms of the CAS motif, at least it is partially attributed to its participation of anchoring GDOWN1 to the cytoplasmic filament subcomplex of the NPC. NPCs are composed of ~32 conserved nucleoporin proteins. Besides their central role as nucleocytoplasmic conduits, recent studies have revealed that Nups play an important role in the maintenance of cellular

homeostasis through their participation in many cellular activities such as chromatin organization, transcription regulation, DNA damage repair, genome stabilization, and cell cycle control, etc. (*Raices and D'Angelo, 2022*). Therefore, our results support the potential involvement of NPCs in recruitment of GDOWN1 to the nuclear periphery and the resultant cytoplasmic retention, suggesting that the nuclear periphery might be the main workplace for GDOWN1 to execute its cytoplasmic functions. When CAS is fully functional, it sufficiently locks GDOWN1 in the cytoplasm so that the function of NES becomes a backup, which explains the phenomenon that GDOWN1 is insensitive to LMB treatment under this circumstance. Thus, our data suggest that removing or at least alleviating the constraint of CAS would be a prerequisite for licensing GDOWN1's nuclear translocation and the following transcription regulatory activities. Besides the NPC-anchoring activity, other working mechanisms of the CAS-directed cytoplasmic retention remain to be explored. In addition, the controlling mechanisms for switching off the CAS activity remain unclear. Based on our findings, one reasonable hypothesis is that post translational modifications of the core arginines within CAS or possibly other amino acids nearby might facilitate this switch via causing a conformational change or affecting the interactions of GDOWN1 to its regulatory factors (illustrated in *Figure 7D*), which is similar to the reported cases in the literature (*Ashida et al., 2022*; *Navarro-Lérida et al., 2021*).

Our data demonstrate that mutation of the CAS motif immediately switches GDOWN1 into an LMB-sensitive nucleocytoplasmic shuttling protein, and its nuclear abundance is determined by the dynamic balance between its functionally associated binding partners (such as Pol II) and the CRM1/RAN-mediated nuclear export machinery. This partial translocation of GDOWN1 leads to tremendous changes inside of the nucleus, including the reduction of Pol II and the global transcriptional decrease. The less Pol II, the less active transcription there is, and vice versa, and this mutual feedback causes the drastic decline of the cellular transcription levels. In the previously published *in vitro* biochemical data, we and others found that GDOWN1 strongly inhibited both the transcriptional initiation (*Jishage et al., 2012*) and elongation (*Cheng et al., 2012*). In this study, our data indicated that the nuclear GDOWN1 tended to affect the S5P-Pol II more than the S2P subgroup, suggesting that GDOWN1 may have a stronger or a more rapid effect on the transcription initiation than elongation. It's also worth noting that the effects of transcription inhibition and pol II defect may be attributed to the overexpression and massive nuclear accumulation of the GDOWN1(*10M*) mutant, although its expression level in our stable cell line was only about one-fold higher compared with the endogenous GDOWN1 (*Figure 5—figure supplement 1A*). Certainly, further studies are required to figure out the working mechanisms of the nuclear GDOWN1. Moreover, it was suggested that GDOWN1 was involved in the cytoplasmic assembly of Pol II as well (*Ball et al., 2022*; *Forget et al., 2010*; *Forget et al., 2013*). Therefore, the nuclear translocation and accumulation of GDOWN1 may also lead to the reduced efficiency of both Pol II assembly and the following nuclear import, which contributes the observed global transcriptional repression.

Recently it was reported that GDOWN1 played a role in facilitating global transcriptional shut down during mitosis and the genetic ablation of *GDOWN1* exhibited mitotic defects (*Ball et al., 2022*), which is consistent with GDOWN1's stringent localization in the cytoplasm during the interphase. Our discovery of GDOWN1's nuclear translocation upon cellular stresses further expands the context in which GDOWN1 plays an essential role in the global transcriptional repression. The cells without GDOWN1 are much more sensitive to the cellular stresses, emphasizing that GDOWN1 is a crucial factor in maintaining cellular homeostasis. Further studies are needed to explore GDOWN1's functions in the cytoplasm and to identify more cellular situations that may trigger its nuclear translocation. Overall, this work uncovered GDOWN1's new functions and switchable localization in mammalian somatic cells and shed a light on a new connection between the global transcriptional regulation and the cellular stress adaptation.

## Materials and methods

**Key resources table**

| Reagent type (species) or resource | Designation | Source or reference | Identifiers | Additional information |
|---|---|---|---|---|
| Gene (human) | *POLR2M* | EMBL database | ENST00000299638.8 | RNA polymerase II subunit M (GDOWN1) |

*Continued on next page*

| Reagent type (species) or resource | Designation | Source or reference | Identifiers | Additional information |
|---|---|---|---|---|
| Gene (human) | POLR2E | EMBL database | ENST00000615234.5 | RNA polymerase II subunit E (RPB5) |
| Gene (human) | NELFA | EMBL database | ENST00000382882.9 | negative elongation factor complex member A. Provided by Dr. Ruichuan Chen |
| Gene (human) | NELFE | EMBL database | ENST00000375429.8 | negative elongation factor complex member E. Provided by Dr. Ruichuan Chen |
| Gene (human) | SUPT4H1 | EMBL database | ENST00000225504.8 | SPT4 homolog, DSIF elongation factor subunit. |
| Gene (human) | SUPT5H1 | EMBL database | ENST00000599117.5 | SPT5 homolog, DSIF elongation factor subunit |
| Gene (human) | MED1 | EMBL database | ENST00000300651.11 | mediator complex subunit 1. Provided by Dr. Ruichuan Chen |
| Gene (human) | MED26 | EMBL database | ENST00000263390.8 | mediator complex subunit 26. Provided by Dr. Ruichuan Chen |
| Gene (human) | RPRD1A | EMBL database | ENST00000399022.9 | regulation of nuclear pre-mRNA domain containing 1 A |
| Gene (human) | RPRD1B | EMBL database | ENST00000373433.9 | regulation of nuclear pre-mRNA domain containing 1B |
| Gene (human) | VDAC1 | EMBL database | ENST00000265333.8 | Voltage-dependent anion-selective channel protein 1 |
| Gene (human) | GALNT2 | EMBL database | ENST00000366672.5 | Polypeptide N-acetyl-galactosaminyl transferase 2 |
| Gene (human) | PDIA3 | EMBL database | ENST00000300289.10 | Protein disulfide-isomerase A3 |
| Gene (fly) | Gdown1 | NCBI database | NM_142537.2 | Gdown1 of Drosophila melanogaster. cDNA provided from Mr. Bingtao Niu |
| Gene (zebrafish) | Gdown1 (Polr2m) | NCBI database | NM_001346180.1 | Gdown1 of Danio rerio. cDNA provided from Dr. Yingmei Zhang |
| Cell line (Homo-sapiens) | HeLa | National Collection of Authenticated Cell Cultures | TCHu187 | Authenticated by STR profiling; free from mycoplasma and other microorganisms |
| Cell line (Homo-sapiens) | HEK293T | National Collection of Authenticated Cell Cultures | GNHu17 | Authenticated by STR profiling; free from mycoplasma and other microorganisms |
| Cell line (Homo-sapiens) | GES-1 | | | Cell line maintained in Kesheng Li lab; Authenticated by STR profiling; free from mycoplasma and other microorganisms |
| Cell line (Homo-sapiens) | MKN45 | Provided by Dr. Kesheng Li, Gansu Provincial Academic Institute for Medical Research, China | | Cell line maintained in Kesheng Li lab; Authenticated by STR profiling; free from mycoplasma and other microorganisms |
| Cell line (Homo-sapiens) | SW620 | National Collection of Authenticated Cell Cultures | TCHu101 | Authenticated by STR profiling; free from mycoplasma and other microorganisms |
| Cell line (M. musculus) | NIH3T3 | National Collection of Authenticated Cell Cultures | GNM 6 | Authenticated by STR profiling; free from mycoplasma and other microorganisms |
| Cell line (M. musculus) | E14TG2a | Provided by Dr. Qintong Li, Sichuan university, China. | | Originally purchased from ATCC, further adapted to be feeder-free. Authenticated by STR profiling; free from mycoplasma and other microorganisms |

| Reagent type (species) or resource | Designation | Source or reference | Identifiers | Additional information |
|---|---|---|---|---|
| Recombinant DNA reagent | pBiFC (VN- or YC-) (plasmid) | Provided by Dr. Tom Kerppola, University of Michigan Medical School, USA | | Plasmids used in BiFC assay |
| Recombinant DNA reagent | pTripZ (plasmid) | Addgene | #127696 | Lentiviral vector for inducible expression in mammalian cells |
| Recombinant DNA reagent | pMD2.G (plasmid) | Addgene | #12259 | Lentivirus packaging vector |
| Recombinant DNA reagent | psPAX2 (plasmid) | Addgene | #12260 | Lentivirus packaging vector |
| Recombinant DNA reagent | pcDNA3.1(+) (plasmid) | Addgene | #78110 | Gene expression vector |
| Recombinant DNA reagent | pX459 (plasmid) | Addgene | #118632 | CRISPR-Cas9 vector |
| Antibody | anti-GDOWN1 (Rabbit polyclonal) | In this study, generated in Biodragon, Suzhou, China | | Immunogen: human GDOWN1 (251–368 aa); WB: 1:1000 Preferably used in this study without further indication. |
| Antibody | anti-GDOWN1 (Sheep polyclonal) | Provided by Dr. David Price, The University of Iowa, USA | | Immunogen: human GDOWN1 (full length); WB: 1:1000 |
| Antibody | anti-α-TUBULIN (Mouse monoclonal) | Biodragon, Suzhou, China | Cat# B1052 | WB: 1:10000 |
| Antibody | anti-FBL/Fibrillarin (Rabbit monoclonal) | Abclonal, Wuhan, China | Cat# A0850 | Nucleoli marker WB: 1:10000 |
| Antibody | anti-CRM1/XPO1 (Rabbit polyclonal) | Abclonal, Wuhan, China | Cat# A0299 | WB: 1:1000 |
| Antibody | anti-RAN (Rabbit polyclonal) | Abclonal, Wuhan, China | Cat# A0976 | WB: 1:1000 |
| Antibody | anti-RPB1-total (Rabbit monoclonal) | CST, Massachusetts, USA | Cat# 14958 S Clone D8L4Y | Immunogen: a synthetic peptide corresponding to the residues surrounding N613 of the human RPB1; WB: 1:1000 |
| Antibody | anti-RPB1-unphosphorylated (Mouse monoclonal) | Abcam, Boston, USA | Cat# AB817 Clone 8WG16 | Immunogen: the purified wheat germ Pol II; IF: 1:200; WB: 1:1000 |
| Antibody | anti-RPB1-Ser5-Phos (Mouse monoclonal) | BioLegend, California, USA | Cat# 904001 Clone CTD4H8 | Immunogen: a peptide containing 10 repeats of the synthetic peptide YSPTSPS with S5 positions phosphorylated; IF: 1:1000 |
| Antibody | anti-RPB1-Ser2-Phos (Rabbit polyclonal) | Abcam, Boston, USA | Cat# AB5095 | Immunogen: a peptide of the CTD repeats of YSPTSPS from *S. cerevisiae* Pol II with S2 positions phosphorylated; IF: 1:200 |
| Antibody | anti-H3 (Mouse monoclonal) | Biodragon, Suzhou, China | Cat# B1055 Clone 1G1 | WB: 1:500000 |
| Antibody | anti-G3BP1 (Rabbit monoclonal) | Abclonal, Wuhan, China | Cat# A3968 | IF: 1:500 |
| Antibody | anti-Flag (Mouse monoclonal) | Abmart, Shanghai, China | Cat# M20008 Clone 3B9 | WB: 1:2000 IF: 1:300 IP: 1:500 |
| Antibody | HRP-conjugated goat-anti-rabbit IgG (Goat polyclonal) | Biodragon, Suzhou, China | Cat# BF03008 | WB: 1:10000 |

| Reagent type (species) or resource | Designation | Source or reference | Identifiers | Additional information |
|---|---|---|---|---|
| Antibody | HRP-conjugated goat-anti-mouse IgG (Goat polyclonal) | Biodragon, Suzhou, China | Cat# BF03001 | WB: 1:10000 |
| Antibody | HRP-conjugated goat-anti-sheep IgG (Goat polyclonal) | Biodragon, Suzhou, China | Cat# BF03025 | WB: 1:10000 |
| Antibody | Goat-anti-mouse IgG/Alexa Fluor 594 (Goat polyclonal) | Abcam, Boston, USA | Cat# AB150116 | IF: 1:200 |
| Antibody | Goat-anti-Rabbit IgG/Alexa Fluor 594 (Goat polyclonal) | Abcam, Boston, USA | Cat# AB150080 | IF: 1:200 |
| Chemical compound, drug | Leptomycin B (LMB) | Beyotime, Shanghai, China | Cat# S1726-10 | |
| Chemical compound, drug | Doxycycline (Dox) | Biogems, California, USA | Cat# 2431450 | |
| Chemical compound, drug | Puromycin | InvivoGen, USA | Cat# ant-pr-1 | |
| Chemical compound, drug | $NaAsO_2$ | INNOCHEM, Beijing, China | Cat# A25410 | |
| Chemical compound, drug | Hoechst 33342 | Solarbio Life Sciences, Beijing, China | Cat# C0031 | |
| Chemical compound, drug | Camptothecin (CPT) | Selleck, Houston, USA | Cat# S1288 | |
| Chemical compound, drug | Doxorubicin hydrochloride | Sangon Biotech, Shanghai, China | Cat# A603456 | |
| Chemical compound, drug | Cycloheximide (CHX) | MedChemExpress, New Jersey, USA | Cat# HY-12320 | |
| | Madrasin | | Cat# HY-100236 | |
| | Tubercidin | | Cat# HY-100126 | |
| Chemical compound, drug | 5, 6-dichloro-1-β-D-ribofuranosylbenzimidazole (DRB) | Sigma, USA | Cat# D1916 | |
| Chemical compound, drug | Propidium iodide (PI) | Solarbio Life Sciences, Beijing, China | Cat# C0080 | |
| Commercial assay or kit | Cell-Light EU Apollo643 RNA Imaging Kit | RIBOBIO, Guangzhou, China | Cat# C10316-2 | |
| Software, algorithm | ImageJ | NIH | | Image analysis |
| Software, algorithm | GraphPad Prism 8.0.2 | GraphPad Software | | Data analysis |
| Software, algorithm | Gen5 | Cytation 5 | | Data acquiring and analysis |
| Software, algorithm | RTCA Software Lite | RTCA | | Data acquiring and analysis |
| Software, algorithm | NIS-ELEMENTS C | Nikon confocal microscope | | Data acquiring and analysis |
| Software, algorithm | BD software | BD LSRFortessa | | Data acquiring |
| Software, algorithm | FlowJQ v10 software | https://www.bdbiosciences.com/zh-cn/products/software/flowjo-v10-software | | Data analysis |
| Software, algorithm | ChopChop | http://chopchop.cbu.uib.no/ | | sgRNA design |
| Software, algorithm | AlphaFold Protein Structure Database | https://alphafold.ebi.ac.uk/ | | Structural prediction of GDOWN1 |
| Software, algorithm | PONDR | http://www.pondr.com/ | | Prediction of the natural disordered regions of human GDOWN1 |

| Reagent type (species) or resource | Designation | Source or reference | Identifiers | Additional information |
|---|---|---|---|---|
| Software, algorithm | The CUCKOO Workgroup | http://msp.biocuckoo.org/online.php | | Prediction of arginine methylation of human GDOWN1 |
| Other | Exfect Transfection Reagent | Vazyme, Nanjing, China | Cat# T101-02 | This reagent was used in the transfection experiments throughout this article. |

## Cell culture, transfection, and drug treatment

HeLa cells and all the other cell lines except for E14Tg2a were cultured in Dulbecco's Modified Eagle's Media (12800–017, Gibco, New York, USA) supplemented with 10% Newborn Calf Serum (04-102-1A, Biological Industries, Haemek, Israel) and pen/strep. The mouse embryonic stem cell line, E14Tg2a, (gift of Dr. Qintong Li in Sichuan University) was cultured in Dulbecco's Modified Eagle's Media supplemented with 15% Fetal Bovine Serum (900–108, Gemini Bio-products, California, USA), 1 x non-essential amino acids (11140–035, Gibco, New York, USA), 200 mM L-glutamine, 0.1 mM β-mercaptoethanol, $10^3$ U/mL leukemia inhibitory factor (LIF, purified in lab), and pen/strep. All the dishes or coverslips used for culturing E14Tg2a cells were pretreated with 0.5% gelatin. All cells were maintained at 37°C, 90% humidity, and 5% CO2. Plasmid transfections were carried out using the Xfect Transfection Reagent according to the manufacturer's protocol. 0.25 µg plasmid was used for transfecting one well of cells in a 24-well cell culture dish and normally confocal microscopy images were taken at 24 hr post transfection.

For samples treated with LMB, 20 nM final concentration of LMB was added to the culture medium at 18 hr post transfection and incubated for 6 hr before data collection (or mock treated with an equal volume of ethanol). For $NaA_sO_2$, DRB, CHX, Madrasin, Tubercidin, CPT, and Doxorubicin treatment, the drug was added to the complete medium at the indicated final concentration and incubated with cells for the indicated timing. Cells were washes for three times with PBS to remove the drug before further operations were pursued.

## Construction of plasmids and stable cell lines

### Plasmid table

| Names of Plasmids | Mutant type and position(s) of mutagenesis (if applicable) | Corresponding figure No. |
|---|---|---|
| pCDNA3.1-GOWN1-Venus | | 1 A, 4 A |
| pBiFC-Flag-cherry-GDOWN1 | | 1A |
| pBiFC-Flag-GDOWN1 | | 1A |
| pBiFC-Flag-NLS-GDOWN1-NLS | | 1A |
| pX459-sg#1 for GDOWN1 | | 1-S1A, S1B |
| pX459-sg#7 for GDOWN1 | | 1-S1A, S1B, 7B, 7D, 7-S1B |
| pX459-sg#8 for GDOWN1 | | 1B, 1-S1A, 7B, 7D, 7-S1B |
| pX459-sg#9 for GDOWN1 | | 1-S1A, S1B |
| pX459-sg#10 for GDOWN1 | | 1-S1A, S1B, 7D |
| pBiFC-Flag-VN-GDOWN1 | | 1D, 1E, 1-S1D, 3D |
| pBiFC-Flag-YC-RPB5 | | 1D |
| pBiFC-Flag-YC-SPT4 | | 1D, 1E |
| pBiFC-Flag-YC-RPRD1A | | 1D |

*Continued on next page*

*Continued*

| Names of Plasmids | Mutant type and position(s) of mutagenesis (if applicable) | Corresponding figure No. |
|---|---|---|
| pBiFC-Flag-YC-MED1 | | 1D |
| pBiFC-Flag-YC-NELE-E | | 1D, 1E, 2B, 2-S1B, 3B, 3-S1D, 4-S1D |
| pBiFC-Flag-YC-SPT5 | | 1D |
| pBiFC-Flag-YC-RPRD1B | | 1D |
| pBiFC-Flag-YC-MED26 | | 1D |
| pBiFC-Flag-VN-NELF-A | | 1E |
| pBiFC-Flag-VN-SPT5 | | 1E |
| pBiFC-Flag-YC-GDOWN1 | | 1D, 1E, 2 C |
| pBiFC-Flag-YC-3x NLS$^{RYBP}$-GDOWN1 | | 1E |
| pBiFC-Flag-VN-3x NLS$^{RYBP}$-GDOWN1 | | 1E |
| pBiFC-Flag-YC-VDAC1 | | 1-S1D |
| pBiFC-Flag-YC-GALNT2 | | 1-S1D |
| pBiFC-Flag-YC-PDIA3 | | 1-S1D |
| pBiFC-Flag-VN-GDOWN1(*m1*) | A truncated mutant (1–110 aa) | 2B |
| pBiFC-Flag-VN-GDOWN1(*m2*) | A truncated mutant (111–250 aa) | 2B |
| pBiFC-Flag-VN-GDOWN1(*m3*) | A truncated mutant (251–368 aa) | 2B |
| pBiFC-Flag-VN-GDOWN1(*m4*) | A truncated mutant (1–250 aa) | 2B |
| pBiFC-Flag-VN-GDOWN1(*m4\**) | A truncated mutant with the NES1 motif mutated [1–250 aa (L191, F192, I193, I199 to Gs)] | 2B |
| pBiFC-Flag-VN-CRM1 | | 2C |
| pBiFC-Flag-VN-RAN | | 2C |
| pTripZ-GDOWN1-Venus-Flag | | 2-S1A, 3 C, 3E, 3-S2A, 5, 5-S1 |
| pTripZ-GDOWN1(*m1*)-Venus-Flag | A truncated mutant (1–110 aa) | 2-S1A |
| pTripZ-GDOWN1(*m2*)-Venus-Flag | A truncated mutant (111–250 aa) | 2-S1A |
| pTripZ-GDOWN1(*m3*)-Venus-Flag | A truncated mutant (251–368 aa) | 2-S1A |
| pBiFC-Flag-VN-GDOWN1(*m5*) | A truncated mutant (1–145 aa) | 2-S1B |
| pBiFC-Flag-VN-GDOWN1(*m6*) | A truncated mutant (1–180 aa) | 2-S1B |
| pBiFC-Flag-VN-GDOWN1(*m7*) | A truncated mutant (1–215 aa) | 2-S1B |

*Continued*

| Names of Plasmids | Mutant type and position(s) of mutagenesis (if applicable) | Corresponding figure No. |
|---|---|---|
| pBiFC-Flag-VN-GDOWN1(*m8*) | A truncated mutant (1–110+251-368 aa) | 3B, 3-S1A |
| pBiFC-Flag-VN-GDOWN1(*m9*) | A truncated mutant (1–110+251-340 aa) | 3B, 3-S1A |
| pBiFC-Flag-VN-GDOWN1(*m9\**) | A truncated mutant with the NES2 motif mutated [1–110+251-340 aa (L332, L336, I338 to Gs)] | 3B, 3-S1A |
| pBiFC-Flag-VN-GDOWN1(*m10*) | A deletion mutant (delete 352–361 aa) | 3B, 3-S1A |
| pBiFC-Flag-VN-GDOWN1(*m11*) | A three-point mutant in CAS motif (R352, R354, R357 to As) | 3B, 3D, 3-S1A |
| pTripZ-GDOWN1(*m11*)-Venus-Flag | A three-point mutant in CAS motif (R352, R354, R357 to As) | 3 C, 3E, 3-S2A |
| pBiFC-Flag-YC-RAE1 | | 3D |
| pBiFC-Flag-YC-NUP50 | | 3D |
| pBiFC-Flag-VN-GDOWN1(*m12*) | A truncated mutant (1–110+251-280 aa) | 3-S1D |
| pBiFC-Flag-VN-GDOWN1(*m13*) | A truncated mutant (1–110+251-310 aa) | 3-S1D |
| pBiFC-Flag-VN-GDOWN1(*m14*) | A deletion mutant (delete 366–368 aa) | 3-S1D |
| pBiFC-Flag-VN-GDOWN1(*m15*) | A deletion mutant (delete 361–368 aa) | 3-S1D |
| pBiFC-Flag-VN-GDOWN1(*m16*) | A deletion mutant (delete 356–368 aa) | 3-S1D |
| pBiFC-Flag-VN-GDOWN1(*m17*) | A three-point mutant (E355, V356, D358 to As) | 3-S1D |
| pBiFC-Flag-VN-GDOWN1(*m18*) | A four-point mutant (D358, E359, D360 D361 to As) | 3-S1D |
| pCDNA3.1-GDOWN1(*mNES2 +CAS*)-Venus | A six-point mutant in NES2 and CAS motifs (L332, L336, I338 to Gs and R352, R354, R357 to As) | 4A |
| pCDNA3.1-GDOWN1(*mNES1*)-Venus | A four-point mutant in NES1 motif (L191, F192, I193, I199 to Gs) | 4A |
| pCDNA3.1-GDOWN1(*mNES1 +CAS*)-Venus | A seven-point mutant in NES1 and CAS motifs (L191, F192, I193, I199 to Gs and R352, R354, R357 to As) | 4A |
| pCDNA3.1-GDOWN1(*mNES2*)-Venus | A three-point mutant in NES2 motif (L332, L336, I338 to Gs) | 4A |
| pCDNA3.1-GDOWN1(*mNES1 +2*)-Venus | A seven-point mutant in NES1 and NES2 motifs (L191, F192, I193, I199, L332, L336, I338 to Gs) | 4A |
| pCDNA3.1-GDOWN1(*mCAS*)-Venus | A three-point mutant in CAS motif (R352, R354, R357 to As) | 4A |
| pTripZ-GDOWN1(*mNES2*)-Venus-Flag | A three-point mutant in NES2 motif (L332, L336, I338 to Gs) | 4C |
| pBiFC-Flag-VN-fGdown1 | The wild type Gdown1 of *Drosophila melanogaster* | 4D, 4-S1D |
| pBiFC-Flag-VN-fGdown1(R325, 328 A) | A double-point mutant in CAS motif (R325, R328A) | 4D, 4-S1D |

*Continued on next page*

*Continued*

| Names of Plasmids | Mutant type and position(s) of mutagenesis (if applicable) | Corresponding figure No. |
|---|---|---|
| pBiFC-Flag-VN-zGdown1 | The wild type Gdown1 of *Danio rerio* | 4D, 4-S1D |
| pBiFC-Flag-VN-zGdown1(R328A, H331A) | A double-point mutant in CAS motif (R328A, H331A) | 4D, 4-S1D |
| pTripZ-GDOWN1(*10M*)-Venus-Flag | A ten-point mutant in NES1, NES2 and CAS motifs (L191, F192, I193, I199, L332, L336, I338 to Gs and R352, R354, R357 to As) | 5, 5-S1 |
| pTripZ-NLS$^{SV40}$-GDOWN1-Venus-Flag | | 5, 5-S1 |
| pTripZ-NLS$^{SV40}$-GDOWN1(*10M*)-Venus-Flag | A ten-point mutant in NES1, NES2 and CAS motifs (L191, F192, I193, I199, L332, L336, I338 to Gs and R352, R354, R357 to As) | 5, 5-S1 |

The pBiFC-Flag-VN (aa 1–172 of Venus) or pBiFC-Flag-YC (aa 173–238 of YFP) plasmids (gifts from Dr. Tom Kerppola, University of Michigan) were used as the parental vectors for generating all the indicated BiFC plasmids. The coding sequences of human *MED1*, *MED26*, and *SPT5* genes were PCR amplified from the plasmids (gifts from Dr. Ruichuan Chen, Xiamen University, *Lu et al., 2016*). Human *GDOWN1* (also namely *POLR2M*, NM_015532.5) and other genes were all amplified by RT-PCR used cDNA samples generated from HeLa cells as the templates. The *Gdown1* genes in *Danio rerio* (NM_001346180.1) and *Drosophila melanogaster* (NM_142537.2) were cloned from the cDNA samples generated directly from animal lysates. Total RNA was extracted by the MolPure Cell/Tissue Total RNA Kit (19221ES50, YEASEN, Shanghai, China) and the cDNA was synthesized using the 1st Strand cDNA Synthesis SuperMix (11141ES60, YEASEN, Shanghai, China). The purified RT-PCR products were double digested by BamHI (R3136S, NEB, Massachusetts, USA) and XbaI (R0145S, NEB, Massachusetts, USA) and then ligated into pBiFC-Flag-VN or -YC vectors by T4 DNA ligase, or when these two restriction enzymes had cut sites within the cDNA sequences, the PCR products were assembled into pBiFC-Flag-VN or -YC vectors via homologous recombination using the ClonExpressII One Step Cloning Kit (C112-02, Vazyme, Nanjing, China). The two NLS motifs in a plasmid named Flag-NLS-GDOWN1-NLS in *Figure 1A* were adopted from the pX459 plasmid originally constructed from Dr. Feng Zhang's lab in MIT (*Ran et al., 2013*). The three NLS motifs in the VN/YC-3xNLS-GDOWN1 plasmids shown in *Figure 1E* were cloned from the CDS sequences of human *RYBP* gene corresponding to the amino acids 1–94 (*Tan et al., 2017*). The truncated fragments of human *GDOWN1* were amplified using the full-length CDS as a template and further used to construct the pBiFC-based GDOWN1 mutants. Point mutations were introduced by designing the long PCR primers containing the designated mutated sequences and then amplified the fragments in the regular PCR or bridging PCR reactions as needed. The information on the amino acids for mutagenesis was shown in *Figures 2A and 3A*. The above pBiFC-based plasmids series were applied in both BiFC assays (directly monitoring the BiFC signals) or in IF assays (detection via a Flag antibody) as indicated in the figure legends. For generating the GDOWN1-Venus plasmid series, pcDNA3.1(+) was used as a parental vector. The full-length, wild type GDOWN1 was amplified from the above pBiFC vector and ligated into the pcDNA3.1-Venus plasmid (previously constructed in lab). The *NES* and/or *CAS* mutant fragments were PCR amplified from the above pBiFC plasmids expressing the corresponding mutant GDOWN1 and further amplified by the bridging PCR and then assembled into the pcDNA3.1-Venus plasmid.

*GDOWN1 KO* HeLa cells were generated via the CRISPR-Cas9 technology. The sgRNAs were selected according to the information provided by ChopChop. The targeting sequences of sgRNAs are listed in the table down below. The pX459-sg*GDOWN1*-#1/#7/#8/#9/#10 plasmids were constructed and transfected into HeLa cells and the cells were selected with 0.5 μg/mL puromycin starting from 48 hr post transfection. After 5 days of selection, the survived cells were pooled and further verified by sequencing and WB. For cells transfected with pX459-sg*GDOWN1*-#1, the pooled cells after puromycin selection were re-plated in a p100 cell culture dish at a density of 2000 cells per dish. After 15 days of culture, the single colonies were picked to a 96-well plate and further expanded. Genomic DNA was isolated and PCR amplified using the primers shown in the table for the verification. The PCR products were gel purified and sent for sequencing (Tsingke Biotechnology, Beijing, China).

| sgRNA# | Targeting sequences of sgRNA | PCR verification primers (*Forward*; *Reverse*) |
| --- | --- | --- |
| 1 | GCGGGAAATGTTGAAGCGCC | GCATGAATGCTCACACAAGG; CGAATGTGACTGAGTCAAAGT |
| 7 | ACGAGTAAGCTGGGGTCCCG | |
| 8 | GTTACAGAGGATCACCATTG | CAGAATTCTGACCCGATAC; CTTCCCACCTCAGCCTCCTGAG |
| 9 | AACTTGACAGGCCTTTCCAG | |
| 10 | TTGATGACATCACAGCAGCT | GGAGGAGAATTAATTGCTAAG; GCAGTTCTAGCAACTTTGTG |

For generating the HeLa cell lines stably expressing GDOWN1, pTripZ, the lentiviral expression vector, was used as a parental vector and the fragment, Venus-Flag, was initially inserted into the pTripZ empty vector to replace the original shRNA expression cassette. The wild type or mutant GDOWN1 fragments were PCR amplified from the constructed pcDNA3.1-based plasmids (for the ones with the NLS addition, the sequences of NLS$^{SV40}$ were attached to the N-terminus of the corresponding primers) and were further inserted in between the TRE-CMV promoter and the Venus gene to obtain iGDOWN1-Venus-Flag plasmids ('i' stands for inducible). For the viral packaging procedure, HEK293T cells cultured in a 6 well plate were transfected with 1 µg pMD2G, 2 µg pAX2, and 3 µg pTripZ-GDOWN1-Venus (WT or mutant) and medium was refreshed at 6 hr post transfection. The viral stock was harvested after 72 hr and further infected HeLa cells for 12 hr. The cells were recovered for one day and further subjected for the puromycin selection (0.5 µg/mL) for 14 days. The surviving cells were pooled and the inducible expression of the GDOWN1-Venus-Flag proteins was verified by WB with a Flag antibody after adding 2.5 µg/mL of Dox for 12 hr.

## BiFC assays

For BiFC assays, HeLa cells were grown on coverslips in 24-well cell culture dishes and 0.25 µg of each pBiFC plasmid (VN- or YC-) was used for co-transfection per well. At 24 hr post transfection, the cells were fixed with 4% formaldehyde for 20 minutes at the room temperature, washed with PBS, stained with 1 µg/mL of Hoechst 33342 for 10 minutes, washed with PBS, and finally visualized in PBS.

## Immunofluorescence and the data analyses

For immunofluorescence assays, the cells were grown on coverslips, fixed with 4% formaldehyde for 20 minutes at the room temperature, washed three times with PBS, dehydrated with 90% methanol at –20°C for 30 minutes, permeabilized with 0.5% Triton X-100 at the room temperature for 20 minutes, washed three times with PBS, incubated with 5% BSA for 1 hr at the room temperature. The cells were then incubated with a Flag antibody (1:200 diluted in TBST) for 12 hr at 4°C. After being washed for three times with TBST, the cells were subjected for a secondary antibody incubation for 1 hr at the room temperature. The cells were further stained with 1 µg/mL of Hoechst 33342 for 10 minutes, washed with PBS, and visualized in PBS. For data analysis in *Figure 6—figure supplement 1D*, the confocal images were further processed by Gen5. The cellular analysis function of Gen5 was used to calculate the Integral intensity of the Venus signal per cell and the corresponding area and these values were used to define the Venus$^+$ cells from the Venus$^-$ cells. Then the mean signals of the indicated RPB1 form per nucleus (the unphosphorylated, S2P, or S5P) for either the Venus$^+$ or the Venus$^-$ cell population were calculated separately and further processed using Graphpad Prism8.

For the statistical analyses of the stress granules, the cells were grown on coverslips in 24-well cell culture dishes (the inducible cell lines were pre-induced with Dox for one day) and then either mock treated or treated with 0.1 mM of NaAsO$_2$ for 6 hr. Then the cells were washed with PBS for three times and incubated with the G3BP1 antibody in the IF assays as described above. For a large-scale data quantification, the images were also acquired using Cytation 5 (Agilent) using the same set of coverslips and at least four regions of interest (ROIs) were randomly selected, and all of the images were further analyzed using the Gen5 software. The cell count (indicated by the Hoechst 33342 signals), G3BP1 spot number per cell, and the mean fluorescence intensity of each G3BP1 spot (within each ROI) were detected and calculated using the built-in tools in the Gen5 software (automatic cell count, spots count, and the subpopulation analysis). The cell containing more than 1 spot was defined as an SG$^+$ cell.

$$\% \text{ of the SG}^+ \text{ cells} = \frac{\text{The number of the SG}^+ \text{ cells}}{\text{The number of the total cells}} \times 100\%$$

## Co-immunoprecipitation

The cells stably expressed GDOWN1-Venus-Flag in a p100 dish were lysed with 500 µL of the lysis buffer (20 mM Tris-HCl, pH 8.0, 150 mM NaCl, 0.5 mM EDTA, 1% NP40, 1% Triton X-100, 50 mM NaF, 1 mM $Na_3VO_4$ [activated], 0.1 mM PMSF, Protease Inhibitor Cocktail [B14012, Bimake, Shanghai, China]) and incubated for 30 minutes at 4°C on a rotator. The lysate was incubated with 25 µL of the anti-Flag magnetic beads (B26101, Bimake, Shanghai, China) for 12 hr at 4°C on a rotator. The beads were washed five times with lysis buffer, and then resuspended with a 5 x loading dye (250 mM Tris-HCl, pH 6.8, 10% SDS, 50% glycerinum, 5% β-mercaptoethanol, 0.1% bromophenol blue) and further pursued for SDS-PAGE analyses.

## Confocal microscopy and the nucleocytoplasmic distribution analyses

The confocal images were obtained using a 100 x oil objective (N.A. 1.45) on a Nikon A1$R$+ Ti2-E laser scanning microscope, equipped with a GaAsP Multi Detector Unit. Images were acquired from at least four randomly selected fields using the NIS-ELEMENTS C software. For data quantification of GDOWN1's nucleocytoplasmic distribution, each set of the obtained images was input into the ImageJ software. The intensity of the Venus signal detected from each cell was obtained and considered as the total signal. The signal of Hoechst 33342 was utilized to define the region of each nucleus, and the Venus signal within the border of the nucleus was recorded as the nuclear signal in the corresponding cell. The cytoplasmic signal can be calculated by subtracting the nuclear signal from the total and the nucleocytoplasmic distribution can be calculated and plotted. For the statistical analyses, the calculated values from each field were averaged and the standard errors (SE) among the four fields were calculated and plotted. The *P* values were calculated via a *t*-test using the built-in tools in Graphpad Prism8.

## EU-Apollo assay and the data analyses

For the EU-Apollo assays, either the parental HeLa cells or the derived stable cell lines were grown on coverslips in 48-well cell culture dishes, and 0.25 µg/mL of Dox was used for the protein induction. 250 µM of EU was added into the culture medium 20 minutes before cell harvest. All the procedures were carried out following the manufacturer's instructions of the Cell-Light EU Apollo 643 RNA Imaging Kit. The confocal images acquired from at least four fields per treatment were used for data analyzed. Based on the fluorescence intensity of the Venus signals, the cells were separated into either Venus$^+$ or Venus$^-$ group. The fluorescence intensity of the EU-Apollo signal was measured cell by cell with ImageJ, and the averaged EU-Apollo signal for each group of the cells per field was calculated and plotted in bar graphs. For the statistical analyses, the averaged values and SEs of the EU-Apollo signals among the four fields were calculated and further processed to obtain the p values via a *t*-test using the built-in tools in Graphpad Prism8.

## Cell fractionation and the quantitative analyses of WB data

About 2x10^6 cells were used for each cell fractionation assay. Freshly harvested cell pellet was resuspended with 100 µL of the cytoplasmic extraction buffer (20 mM Hepes, 1 mM EDTA, 10 mM KCl, 2 mM $MgCl_2$, 0.1% NP-40, 1 mM DTT, 0.1 mM PMSF, Protease Inhibitor Cocktail), and incubated at 4°C for 30 mintes. The cell lysate was centrifuged at 1500 rpm for 3 minutes at 4°C, and the supernatant was saved as the cytoplasmic fraction. The remained nuclei pellet was washed for three times and further resuspended with 100 µL of the cytoplasmic extraction buffer. The resuspended nuclei samples were used as the nuclear fraction (containing both the soluble nucleoplasm and the insoluble chromatin). 5 x protein loading dye was added into the above cytoplasmic (C) and nuclear (N) fractions to generate 1 x samples for SDS-PAGE and WB analyses.

For data quantification, ImageJ was employed to acquire the IntDen value (integral optical density) of each band in the obtained WB images.

$$\text{Relative gray value of the RPB1 (normalized with H3)} = \frac{\text{IntDen (RPB1 in the whole cell lysate)}}{\text{IntDen (H3 in the whole cell lysate)}}$$

## Live cell analyses and data analyses

For live cell analyses, the cells were plated in a 48-well cell culture dish 24 hr before the treatment. For the results shown in *Figure 7D*, the cells were incubated with the complete medium supplemented with 0.2 mM NaAsO$_2$, 0.1 µg/mL Hoechst 33342, and 1 µg/mL PI, then immediately analyzed using Cytation 5. Four ROIs from each well were randomly selected and images were acquired using Gen5 software. The count of total cells (using the Hoechst 33342 signal as an indicator) and the dead cells (using the PI signal as an indicator) in each ROI were calculated using the built-in tools (automatic cell count and subpopulation analysis) in the Gen5 software. The calculated values were further processed using Graphpad Prism8.

$$\text{Relative cell viability (\%)} = \frac{\text{The number of the total cells} - \text{The number of the dead cells}}{\text{The number of the total cells}} \times 100\%$$

For the results shown in *Figure 5B (ii)*, cells were grown in 6-well cell culture dishes and induced by 0.25 µg/mL of Dox for 0, 3, 6, 9 days, respectively. Upon harvest, cells were fixed with 4% formaldehyde for 20 minutes at the room temperature, washed with PBS, and visualized in PBS. Images of at least four randomly selected ROIs from each well were acquired using the Gen5 software. For data analyses, the number of the total cells and the Venus$^+$ cells, and the fluorescence intensity of the Venus$^+$ cells were calculated using the built-in tools in Gen5. The obtained data were further processed in Graphpad Prism8 to export figures.

$$\text{Ratio of the Venus}^+ \text{ cells} = \frac{\text{The number of the Venus}^+ \text{ cells}}{\text{The number of the total cells}}$$

For live cell analyses, using RTCA in *Figure 5B (iv)*, the cells were seeded in an E-plate 16 PET (Agilent), and cultured in the xCELLigence RTCA S16 (Agilent) inside of a cell culture incubator. The cell index values and the slope of their changes were acquired by the RTCA Software Lite along with the cell growth. The obtained values were further processed in Graphpad Prism8 for data export.

## Cell counting and flow cytometry analyses (FACS)

For the results shown in *Figure 5B (iv)*, 6×10^4 cells of each cell line were plated on day 0 and cultured with the complete medium. Two experiments were performed at the same time, with the cells either being seeded in 12-well plates for cell counting, or in 6 well plates for FACS. For generating the growth curve, the cells were incubated with 0.25 µg/mL of Dox for 9 days, and the equal number (6×10^4) of cells from each cell line were re-plated into a fresh cell culture dish every three days (on day 3 and 6). The number of cells was counted on days 3, 6, and 9 using a cell counter (Bodboge, Guangzhou, China), and the averaged value was calculated based on at least three independent readings. The finalized cell counts were calculated based on the averaged values and the distinct dilution coefficient upon each passage.

For FACS analyses, the cells were harvested on days 0, 3, and 9 after Dox induction. To gather the dead cells, the culture medium was centrifuged at 1000 ×g for 5 minutes and the cells at the bottom of the tubes were collected. Then the adherent cells in the plates were trypsinized (0.25% trypsin) and collected via a centrifugation at 800 xg for 5 minutes. Both the attached cells and the dead cells originally from the same sample were combined and resuspend with 500 µL of PBS, and further incubated with DAPI (at a final concentration of 5 µg/mL) at dark for 15 minutes. Then, the cells were immediately handled in the Flow Cytometer (LSRFortessa, BD, New Jersey, USA) for the fluorescence detection. For each sample, a minimum of 10,000 cells were analyzed with the FlowJo 7.6 software. A control sample generated by mixing 2/3 of live cells with 1/3 of the formaldehyde fixed dead cells (indicated as DAPI$^+$ cells) was utilized to facilitate the accurate setting of the gating parameters. The data were exported using the FlowJo 7.6 software.

## Acknowledgements

The authors acknowledge Dr. David Price (the University of Iowa) for providing Gdown1 antibody made in sheep (*Cheng et al., 2012*) and for the helpful comments and discussion from his group; Dr. Ruichuan Chen (Xiamen University), Dr. Qintong Li (Sichuan University), Dr. Yingmei Zhang and Bingtao Niu (Lanzhou University) for providing necessary plasmids, cells and animals; many members of Cheng laboratory for their helpful discussion; Yaojia Wang for her help of revising our model diagram. We also

thank the core facilities in the School of Life Sciences, Lanzhou University for providing high-quality instruments and service for our confocal microscopy experiments. We apologize to our colleagues in transcription field whose work was not discussed adequately owing to space constraints. Funding This work was financially supported by the National Natural Science Foundation of China (31771447, 31471233, 31970624 to Bo Cheng), the Foundation of the Ministry of Education Key Laboratory of Cell Activities and Stress Adaptations grant (lzujbky-2021-kb05 to Bo Cheng), the Fundamental Research Funds for the Central Universities (lzujbky-2020-it16 to Zhanwu Zhu), the Gansu Provincial Outstanding Graduate Student "Innovation Star" Project (2022CXZX-096 to Zhanwu Zhu), and the Gansu Provincial Outstanding Graduate Student "Innovation Star" Project (2021CXZX-107 to Jingjing Liu).

## Additional information

### Funding

| Funder | Grant reference number | Author |
| --- | --- | --- |
| The National Natural Science Foundation of China | 31771447 | Bo Cheng |
| The Gansu Provincial Outstanding Graduate Student "Innovation Star" project | 2021CXZX-107 | Jingjing Liu |
| The Gansu Provincial Outstanding Graduate Student "Innovation Star" Project | 2022CXZX-096 | Zhanwu Zhu |
| The Foundation of the Ministry of Education of China, the Fundamental Research Funds for the Central Universities | lzujbky-2020-it16 | Zhanwu Zhu |
| The Foundation of the Ministry of Education of China, Key Laboratory of Cell Activities and Stress Adaptations Grant | lzujbky-2021-kb05 | Bo Cheng |
| The National Natural Science Foundation of China | 31471233 | Bo Cheng |
| The National Natural Science Foundation of China | 31970624 | Bo Cheng |

The funders had no role in study design, data collection and interpretation, or the decision to submit the work for publication.

### Author contributions

Zhanwu Zhu, Conceptualization, Methodology, Writing – original draft, Project administration; Jingjing Liu, Formal analysis, Investigation, Visualization, Project administration; Huan Feng, Conceptualization, Validation, Visualization, Methodology, Project administration; Yanning Zhang, Ruiqi Huang, Qiaochu Pan, Project administration; Jing Nan, Supervision, Writing - review and editing; Ruidong Miao, Supervision, Validation, Writing - review and editing; Bo Cheng, Conceptualization, Supervision, Funding acquisition, Validation, Investigation, Methodology, Writing – original draft, Writing - review and editing

**Author ORCIDs**
Zhanwu Zhu ![ORCID] http://orcid.org/0000-0001-8494-4348
Bo Cheng ![ORCID] http://orcid.org/0000-0002-7060-1616

**Decision letter and Author response**
Decision letter https://doi.org/10.7554/eLife.79116.sa1
Author response https://doi.org/10.7554/eLife.79116.sa2

## Additional files

### Supplementary files
• MDAR checklist

### Data availability
All data generated or analyzed during this study are included in the manuscript and supporting file; Source Data files have been provided for Figures whenever is necessary.

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
