## [Editor Report]

This important study identifies two distinct nuclear export elements and a strong cytoplasmic anchoring sequence in the GDOWN1 transcription factor that restricts its nuclear import and its ability to inhibit RNA polymerase II transcription. The study shows how this mechanism is modulated in stress conditions that promote GDOWN1 nuclear localization as part of a protective response. This study presents compelling evidence for the role of GDOWN1 in transcriptional regulation and should be of wide general interest.

---

## [Decision Letter]

**Decision letter after peer review:**

Thank you for submitting your article "Lifting the ban on nuclear import activates Gdown1-mediated modulation of global transcription and facilitates adaptation to cellular stresses" for consideration by *eLife*. Your article has been reviewed by 2 peer reviewers, and the evaluation has been overseen by a Reviewing Editor and Kevin Struhl as the Senior Editor. The reviewers have opted to remain anonymous.

The referees agreed that this study should be of general interest to readers of *eLife* following an adequate revision that takes into account the issues raised by the referees.

The authors should pay specific attention to the following points.

1). The BiFC results indicate that GDOWN1 appears to interact with all the tested cytoplasmic proteins and moreover most on the deletion mutants also still interact with their nuclear targets. The authors must provide negative controls to demonstrate the specificity of the interactions detected in this way. The referees also question the utility of these experiments that in any case to not add to the conclusions concerning the sequences that regulate GDOWN1 intra-cellular localization.

2). On page 9 the authors claim that M1 localizes mainly in the nucleus. It is more accurate to say partitions between the cytoplasm and nucleus. If fact there are cells where it seems also to remain cytoplasmic. Similarly, in Figure 3C there are cells on the left of the upper panel that also seem to show diffuse cytoplasmic staining. The authors should perform quantitative image analyses. This has been done in some figures but should be added for all.

3). The manuscript requires a major revision by a native English speaker to correct a large number of grammatical errors.

4). The style of the title could also be changed. Lifting the ban sounds too familiar. For example 'Overcoming cytoplasmic retention of GDOWN1 by the combinatorial action of nuclear export and cytoplasmic anchoring signals modulates transcription and facilitates adaption to cellular stress.

Reviewing editor comments.

1). Figure 3D is rather difficult to visualize, could the authors show a higher magnification?

2). In Figure 4B the *Drosophila* Cas mutant appears to accumulate more in the nucleus that the mutated zebrafish protein. This is rather surprising as given that *Drosophila* is more distantly related than zebrafish, but may reflect the fact that *Drosophila* CAS motif seems better conserved. Can the authors comment on this? Quantification analyses of the images would strengthen the results.

3). In Figure 5B, rather than simply using DAPI in the flow cytometry, can the authors use activated caspase 3 or Annexin V to look for apoptosis? Is apoptosis the mechanism of cell death?

4). In Figure 6B there seems to be a perinuclear staining on the iNLS-GDOWN in the bottom right Venus panel. Can the authors comment on this?

5). Many of the experiments in this study are performed with stable or inducible ectopic expression. It is difficult to assess the expression level of the exogenous proteins versus the endogenous by the immunofluorescence. In the supplemental data, the authors performed immunoblots that often showed the exogenous proteins to be expressed at much higher levels that endogenous. To some extent, this reinforces some of the conclusions when the overexpressed mutants remain cytoplasmic as it indicates the robustness of the retention mechanism. However, the authors should mention the potential caveats that overexpression may introduce.

In addition to the above, the authors should also address the various issues raised by the referees and listed below.

*Reviewer #1 (Recommendations for the authors):*

On p. 3, lines 42-43: The general Pol II transcription factors are listed; TFIIS should not be included in that group.

Figure 1D, for example. The authors use BiFC signals to track Gdown1; these rely on the presence in the cytoplasm of proteins that should be nuclear, like Spt4/5. I found this confusing, and the approach did not seem to add anything to straight detection of tagged Gdown1.

On p. 9, lines 198-199: The authors refer to the structurally flexible regions of Gdown1 without any indication of how that determination was made. They should refer to the structural predictions in Figure S4A.

On p. 14, lines 342-345: It is asserted that the function of the CAS motif depends partially on structural support from NES2. "Structural support" seems like an overstatement- I don't think any particular mechanistic connection can be inferred.

P. 17, first paragraph: Do the authors have any sense for the relative levels of the 10M and NLS^-1^0M versions of Gdown1, post dox exposure, as compared to the endogenous Gdown1?

Figure 6B and S6: The overall effect of Gdown1 access to the nucleus is clear, but more could have been learned with a quantitative comparison of the relative loss of total Pol II, ser5P and ser2P. For example, if ser2 is reduced more than ser5 this could imply an effect on elongation. Probing a panel C blot with the appropriate antibodies could have addressed this point.

I could not understand the result in panel 6C. If nuclear export is blocked by LMB, how can most of HeLa Pol II be cytoplasmic after LMB treatment?

Points in the Discussion:

How is Gdown1 acting in the nucleus? The earlier biochemical work provided a number of possibilities but those aren't investigated here or extensively discussed. Is Gdown1 mostly acting to tie up Pol II so that it cannot make PICs, or is it also/mostly acting downstream of initiation to block elongation? As noted above, determining whether ser5 and ser2 Pol II are affected differently might have shed light on this. Additionally, it would have been useful to know if nuclear Gdown1 is chromatin associated (probably talking to elongation) or free in the nucleoplasm (probably bound to Pol II). The point about effects on elongation seems particularly important when trying to understand how nuclear Gdown1 reduced overall Pol II levels. In particular, if elongation were severely affected polymerases could be stalled for extended periods at difficult-to-traverse locations, provoking the ubiquitination and clearance of those complexes (and thus reducing overall Pol II levels).

Finally, lines 604-607: The authors assert that EU labeling in nucleoli is drastically reduced upon Gdown1 entry into nuclei, showing that Gdown1 interferes with both Pol I and Pol III activity. There is no quantitation provided for the assertion that nucleolar labeling is reduced, but regardless such reduction would not show a direct effect on Pol I (or Pol III) activity- strong reductions in mRNA synthesis should feed back into reductions in rRNA synthesis.

*Reviewer #2 (Recommendations for the authors):*

We have a few suggestions to increase the clarity of this manuscript:

1. The current title is a bit complicated. Can the authors make the title simpler?

2. The authors have some factually incorrect statements in their introduction. They should try to improve the content and presentation of the introduction. Examples:

a. In the introduction (line 57-64), they discuss in vitro work done on GDOWN1, but they do not mention the 2018 structure from Roeder (doi: 10.1038/s41594-018-0118-5). Can the authors please add this paper, as it also clarifies what is known about GDOWN1-Pol II interactions?

b. In the introduction (line 89), they mention ChIP-seq data of Gdown1 and elongating Pol II. Can they please clarify this sentence? It is confusing to understand.

c. Line 38: RNA polymerase II is in fact not the only polymerase that synthesis protein coding genes in eukaryotes (e.g., mitochondrial RNA polymerase also synthesizes protein coding mRNA).

3. The figures are very complicated with many panels. Could the authors consider making more subpanels? For example, Figure 1D is showing two different things; what factors GDOWN1 interacts with, and then various factors interacting with other factors. We recommend separating these figures to increase the clarity of the paper.

4. We have a few suggestions for improving Figure 4A. It is of particular interest to see how NES1, NES2, and then the double NES mutation influences nuclear levels. Could they more directly compare these three mutation-types by putting quantitative analysis of all constructs next to each other?

5. While the evolutionary analysis in Figure 4B+C is exciting, we feel it takes away from the figures purpose. We suggest moving figure 4B+C to the supplementals and then expanding on the analysis of the different NES in the main figure.

6. Please make a cartoon describing that 10M is a triple mutation of all three identified domains in figure 5. We generally to produce an additional supplementary figure panel that summarizes all used constructs in a graphical representation.

7. In Figure 6, the authors provide evidence for a decrease in EU RNA incorporation in the context of the triple mutant. It would strengthen the manuscript if the authors could demonstrate a transcriptional defect by providing data from a complementary technique.

8. Could the authors provide additional quantification of Figure S6B?

9. The authors should also provide error bars for Figure S6C.

10. As the authors are well aware, Pol II antibodies show sub-optimal specificity. Could the authors provide data from additional Pol II antibodies (e.g., S2P, S5P, RPB3) for Figure S6C?

11. In lines 455-456, the authors mention in vitro transcription assays, but do not cite anything. Please cite the study being referred to.

12. We ask the authors to discuss why GDOWN1 apparently interacts with so many transcriptional regulators in the cytoplasm. Is GDOWN1 regulating these factors by interacting with them in the cytoplasm?

13. The Alphafold structure in supplement figure 4 – 1A is difficult to see. Can the authors increase clarity by changing to a white background and focusing more on the regions they are showing? Please also add the same cartoon that is used throughout the paper to further help clarifications.

14. The authors should also carefully check their manuscript for missing prepositions etc.

---

## [Author Response]

The authors should pay specific attention to the following points.1). The BiFC results indicate that GDOWN1 appears to interact with all the tested cytoplasmic proteins and moreover most on the deletion mutants also still interact with their nuclear targets. The authors must provide negative controls to demonstrate the specificity of the interactions detected in this way. The referees also question the utility of these experiments that in any case to not add to the conclusions concerning the sequences that regulate GDOWN1 intra-cellular localization.

Thank you for the Reviewers’ comments! The Reviewers concerned about the interacting specificity between GDOWN1 mutants and NELF-E. We agree this part of data is not very easy to be fully understood. GDOWN1 was reported to be directly interacting to multiple components of Pol II (Hu et al., 2006; Jishage et al., 2018). We also found GDOWN1 played an important role in the cytoplasmic assembly of Pol II (our unpublished data), which led us hypothesize that the GDOWN1 (both the full length and the truncated mutants) & NELF-E interactions seen in our BiFC assays were possibly mediated by other factors such as the Pol II components. The binding specificity of our BiFC results have been well proven. Quite a lot of factors were screened in our BiFC assays and some of them did show negative results. To further strengthen the specificity, we added more negative controls in Figure 1—figure supplement 1D. This new collection of negative controls includes three cytoplasmic proteins, VDAC1 (a mitochondrial protein), GALNT2 (a Golgi protein) and PDIA3 (an endoplasmic reticulum protein). These results, together with the two nuclear factors previously shown in Figure 1D (MED1 and MED26), emphasize that GDOWN1 only interacts to its real binding partners.

We prefer to keep the BiFC data in the manuscript since it is beneficial of using BiFC assays to identify key localization-controlling motifs. The detailed reasons are listed as followed:

1) BiFC assays have the advantages of directly exhibiting the interactions between GDOWN1 and its potential binding partners in live cells. Originally, we employed these assays as a quick way to verify the interaction of GDOWN1 to Pol II components, and meanwhile to test our hypothesis that GDOWN1 might physically interact to transcription elongation factors due to their functional coordination (Cheng et al., 2012). Indeed, the BiFC results indicated very strong signals between GDOWN1 and DSIF (SPT4 and SPT5) or NELF (NELF-E and NELF-A etc.). However, these interactions were unexpectedly present in the cytoplasm. As shown in Figure 1E, the BiFC signals of SPT4•SPT5 interaction, and NELF-E•NELF-A interaction were both present in the nucleus as expected. These results actually fit quite well with the fact that GDOWN1 is primarily present in the cytoplasm. Under the BiFC conditions, a portion of the overexpressed nuclear factors are present in the cytoplasm, so that their interactions to the cytoplasmic GDOWN1 become well captured, while these interactions are typically occurring only in the nucleus upon the appearance of GDOWN1.

2) BiFC system has the advantage to stably capture GDOWN1’s binding partner(s). The irreversibility of BiFC has been considered as its drawback in many circumstances, while in our case, it was actually turned it into an advantage for efficiently identifying any possible and even transient interactions between GDOWN1 and other factors. Our BiFC results well supported the data from our immunofluorescence and cell fractionation assays about GDOWN1’s subcellular localization on its own. In addition, they were able to capture protein/protein interactions between GDOWN1 and other factors, even when they only occurred transiently.

3) BiFC system provides better efficiency and contrast in mutant screening for the key localization motifs of GDOWN1. We observed that when being transiently transfected into the cells, BiFC signals could be well detectable very rapidly (as early as 6 hours post transfection), comparing to expressing GDOWN1 tagged with a fluorescent protein (more than 12 hours). Thus, it allowed us to acquire the data at an early time point, which was especially useful when testing the mutants with a potential of triggering cell death. In addition, since GDOWN1 by itself does not contain any strong NLS, its truncation mutants do NOT enter the nucleus actively even when its cytoplasmic retention signal is chopped off. To overcome this potential problem, we employed NELF-E as a potent nuclear partner of GDOWN1 in our BiFC assays to facilitate GDOWN1’s nucleus entry. This difference is clearly demonstrated in Figure 2B, when data from both IF and BiFC are shown side by side. For example, the *m2* mutant was uniformly localized in both cytoplasm and nucleus in the IF result, while it was strictly present in the nucleus in the BiFC result. Similar examples also include Figure 4D and Figure 4—figure supplement 1D. Overall, this better contrast of BiFC data facilitated our mutant screening and simplified our statistical analyses.

The functional efficiency of those regulatory motifs identified by BiFC assays was further confirmed by a battery of tests including IF assays and direct detections of the mutants via monitoring the fused fluorescent tags. Taken together, we believe that the main conclusions from these BiFC results in the manuscript are solid.

2). On page 9 the authors claim that M1 localizes mainly in the nucleus. It is more accurate to say partitions between the cytoplasm and nucleus. If fact there are cells where it seems also to remain cytoplasmic. Similarly, in Figure 3C there are cells on the left of the upper panel that also seem to show diffuse cytoplasmic staining. The authors should perform quantitative image analyses. This has been done in some figures but should be added for all.

Thank you for your pointing out our inaccurate claim here. We corrected it by rewording it into ‘M1 localizes partially in the nucleus’.

We added the quantitative image analyses for Figure 3—figure supplement 1A,1C and the rest of figures in the revised manuscript, including Figure 2B, Figure

2—figure supplement 1B, 1C, Figure 3—figure supplement 1A, 1B, 1D, Figure 4B, Figure 4D (Figure 4—figure supplement 1C), Figure 4—figure supplement 1D, Figure 6—figure supplement 1C etc. Overall, we think these additions improved the quality of our manuscript, thank you!

3). The manuscript requires a major revision by a native English speaker to correct a large number of grammatical errors.

Thank you for pointing out our weakness of writing in English. We now carefully modified our manuscript and corrected those grammatical mistakes.

4). The style of the title could also be changed. Lifting the ban sounds too familiar. For example 'Overcoming cytoplasmic retention of GDOWN1 by the combinatorial action of nuclear export and cytoplasmic anchoring signals modulates transcription and facilitates adaption to cellular stress.

Thank you for this really good suggestion! This title accurately describes the main findings and the key points of our manuscript while it seems a little long. We shortened this title and decided to use the new title shown below. ‘Overcoming the cytoplasmic retention of GDOWN1 modulates global transcription and facilitates stress adaptation’.

Reviewing editor comments.1). Figure 3D is rather difficult to visualize, could the authors show a higher magnification?

Thank you for pointing out this issue! We think the visualization difficulty for Figure 3D is mainly due to the brightness, not the magnification. The actual intensity of BiFC signals were variable across samples. The main point here is to demonstrate the interactions between GDOWN1 and some representative nuclear pore components, and compare their distinct subcellular localizations qualitatively. Thus, for better demonstration, we adjusted the brightness of the images to make the differences of their subcellular localization easier to be visualized.

2). In Figure 4B the Drosophila Cas mutant appears to accumulate more in the nucleus that the mutated zebrafish protein. This is rather surprising as given that Drosophila is more distantly related than zebrafish, but may reflect the fact that Drosophila CAS motif seems better conserved. Can the authors comment on this? Quantification analyses of the images would strengthen the results.

This is an interesting point! In this study, we identified that the three arginines as a whole in the CAS motif were very important for the cytoplasmic retention of human GDOWN1. However, we did not further narrow it down to dissect the contribution of each individual R. From our data, it appeared that fly CAS played a very similar localization-regulatory role to its human counterpart. We followed the Reviewing editor’s suggestion to quantify these images and provided the data in Figure 4─figure supplement 1C. The quantitative data were consistent with the images that the *Drosophila* CAS mutant was indeed accumulated more in the nucleus than the zebrafish mutant upon LMB treatment.

Our experiments of ectopically expressing either fly or fish Gdown1 in human cells mainly tested the conservation of CAS motifs qualitatively but not quantitatively. Overall, we think the data emphasized that the CAS motifs (and some of the associated arginines) were very conserved from the low animals to mammals. Further experiments are necessary to figure out the accurate evolutionary orders across species.

3). In Figure 5B, rather than simply using DAPI in the flow cytometry, can the authors use activated caspase 3 or Annexin V to look for apoptosis? Is apoptosis the mechanism of cell death?

Thank you for this very valuable suggestion! We have been very interested in the type and the mechanism of cell death upon the nuclear accumulation of GDOWN1 as well. Previously, we transiently transfected a GDOWN1 truncation mutant and observed severe cell death within a few hours upon its nuclear accumulation (Author response image 1). These cells were rapidly dying with the appearance of apoptotic-like bodies. Recently, we carried out Flow cytometry experiments using Annexin-V to check any possible signs of early-stage apoptosis. Two cell lines shown in Figure 5B, iGDOWN1(WT)-Venus expressing in the cytoplasm and iNLS-GDOWN1(*10M*) expressing in the nucleus, were compared in parallel. The two cell lines were incubated with Dox for 3 days to induce the expression of the GDOWN1 proteins and then subjected to Flow cytometry analyses. As seen in Author response image 1, the cells expressing GDOWN1-Venus did not show much Annexin-V^+^ signals (1.49% in Quadrant 4), while the cell expressing NLS-GDOWN1(*10M*) indeed showed increased Annexin-V^+^ signals (8.83%). Due to some loss of dead cells during the cell harvest process, the ratio of Annexin-V^+^ cells might be underestimated. These data suggest that the nuclear GDOWN1 may cause apoptosis to some extent.

**Author response image 1. sa2fig1:** The nuclear accumulation of GDOWN1 may trigger apoptosis. (A) Transient transfection of a human GDOWN1 truncation mutant resulted in cell death within hours post LMB-induced nuclear translocation of GDOWN1. In this case, we observed the appearance of apoptotic body-like extracellular vesicles. (B) Flow cytometry analyses for detection of the apoptotic signals comparing the cells expressing the either the cytoplasmic GDOWN1-Venus [iGDOWN1(WT)-Venus] or the nuclear GDOWN1 [iNLS-GDOWN1(*10M*)-Venus]. The X-axis showed the phosphatidylserine (PS) signals of the outer membrane labeled with APC fluorescence, and the Y-axis marked the π signals.

Although the transient expression of the nuclear GDOWN1 resulted in a severe cell death within hours as mentioned above, in the cell line stably expressing NLS-GDOWN1(*10M*), we observed that a significant portion of these cells were able to survive the constant expression of the nuclear GDOWN1 for several days at least. This cell population actually experienced a constant and slow process of cell death, indicating that some other unknown cell-death mechanisms were also existing along with apoptosis. Since further elucidation of these complicated cell death mechanisms is out of the scope of this manuscript, we decided not to include this piece of data or draw any immature conclusion in this manuscript. Further analyses are currently ongoing in our laboratory.

4). In Figure 6B there seems to be a perinuclear staining on the iNLS-GDOWN in the bottom right Venus panel. Can the authors comment on this?

Thank you for pointing this out! We think this observation was consistent to the rest of the findings in this manuscript quite well and happened exactly as we expected! The massive accumulation of the perinuclear signal of NLS-GDOWN1 was contributed by the combinatory effects of two layers of actions. One was the intrinsic nature of cytoplasmic retention determined by GDOWN1’s CAS motif, the other was the artificial driving force for its accumulation in the nucleus enforced by NLS. Apparently, NLS lost this game, as indicated by the enhanced retention of GDOWN1 at the entrance of the nucleus. Overall, this observation emphasized the unusually potent capability of GDOWN1’s retention in the cytoplasm.

5). Many of the experiments in this study are performed with stable or inducible ectopic expression. It is difficult to assess the expression level of the exogenous proteins versus the endogenous by the immunofluorescence. In the supplemental data, the authors performed immunoblots that often showed the exogenous proteins to be expressed at much higher levels that endogenous. To some extent, this reinforces some of the conclusions when the overexpressed mutants remain cytoplasmic as it indicates the robustness of the retention mechanism. However, the authors should mention the potential caveats that overexpression may introduce.

Thank you for this valuable suggestion! In this revised manuscript, we provided immunoblots to directly compare the expression levels of the ectopically expressed GDOWN1 to their endogenous counterpart (shown in Figure 5─figure supplement 1A) and the result is briefly mentioned in the *RESULTS* session of the manuscript. As seen in this figure, this experimental system often resulted in overexpression of the exogenous protein(s) of interest, as the Reviewing Editor concerned. Theoretically, this might generate non-objective observations and conclusions. Luckily, in our case, the three stable cell lines with GDOWN1 overexpressed (~5 fold or more comparing to the endogenous GDOWN1) did not cause much cell death or other cell growth defects, while the one that triggered severe cell death (iNLS-GDOWN1-*10M*) only expressed the exogenous protein at a level comparable to the endogenous. Thus, the general concerns and the potential caveats about studying overexpressed exogenous proteins did not apply here so that the basic observation and the conclusions would not be interfered much. Overall, we believe that our conclusions stated in the manuscript are objective and solid. We also discuss this point in the *DISCUSSION* session.

In addition to the above, the authors should also address the various issues raised by the referees and listed below.Reviewer #1 (Recommendations for the authors):On p. 3, lines 42-43: The general Pol II transcription factors are listed; TFIIS should not be included in that group.

Thank you for your suggestion! It was our mistake and it is reasonable to distinguish TFIIS from other general transcription factors. This factor is now removed from the original description ‘including general transcription factors (TFIID, TFIIA, TFIIB, TFIIF, TFIIE, TFIIH and TFIIS)’ in the manuscript (line 43).

Figure 1D, for example. The authors use BiFC signals to track Gdown1; these rely on the presence in the cytoplasm of proteins that should be nuclear, like Spt4/5. I found this confusing, and the approach did not seem to add anything to straight detection of tagged Gdown1.

Thank you for your comments! Since both reviewers brought up this question, we provided our answers to this question above in the “Common concerns by the Reviewers” session (Q#1).

On p. 9, lines 198-199: The authors refer to the structurally flexible regions of Gdown1 without any indication of how that determination was made. They should refer to the structural predictions in Figure S4A.

Thank you pointing out the unclear statement in our manuscript! We add a Supplementary figure in Figure 2─figure supplement 1A, providing the secondary structural prediction of the entire human GDOWN1 using PSIPRED

(http://bioinf.cs.ucl.ac.uk/psipred/). The three mutants (*m1-m3*) used in Figure 2 are labeled and as can be seen in this figure, the two positions selected to split GDOWN1 (amino acids 110 and 250) are both located within the coiled region, which we believe to have minimal impact on the overall structure of GDOWN1. We now add a brief description in the main text for clarification (line 205).

On p. 14, lines 342-345: It is asserted that the function of the CAS motif depends partially on structural support from NES2. "Structural support" seems like an overstatement- I don't think any particular mechanistic connection can be inferred.

Thanks for this valuable comment! We agree that more evidence is required in order to draw this conclusion. Our data indicated that the functions of NES2 and CAS are interconnected. Since these two regulatory motifs are very close at the primary structural level, we are even lacking of enough evidence that they are absolutely independent to each other. Therefore, we reword the statement here. ‘The above data from the intrinsic motif analyses demonstrate that each one of the two NES motifs of GDOWN1 acts as an independent CRM1-regulated element and the function of CAS motif depends partially on the existence of NES2.’

P. 17, first paragraph: Do the authors have any sense for the relative levels of the 10M and NLS^-1^0M versions of Gdown1, post dox exposure, as compared to the endogenous Gdown1?

To address reviewer’s concern about the expression levels across the stable cell lines used on Figure 5, we add the immunoblots of these cell lines in Figure 5─figure supplement 1A, showing the expression levels of both endogenous and exogenous GDOWN1 in the presence or absence of Dox induction. As seen in this figure, the expression levels of the exogenous GDOWN1 are all higher than the endogenous counterpart, ranging from a couple to several folds. When comparing the expressions in each cell line pairs, it can be seen that the level of the GDOWN1(*10M*) is similar to that of wild type GDOWN1, while the level of the NLS-GDOWN1(*10M*) is much less than that of NLS-GDOWN1(WT). However, our results from various experiments shown in Figure 5 clearly and consistently indicated that the equal or less expressions of the 10M mutants achieved more severe defects on global transcription and cellular growth. Therefore, we believe that the drastic defects observed in the cells with the 10M mutants being expressed are mainly attributed to the nuclear accumulation and functions of GDOWN1. It is worth to note that the expression level of NLSGDOWN1(*10M*) is at a level very close to that of the endogenous, which is the lowest among the four cell lines, while it exhibits the most severe defects. Thus, our major findings claimed in the manuscript were not generated due to the excessive expression of exogenous GDOWN1.

In the revised manuscript, besides adding the above WB data in the supplementary figure, we also pointed out this results in the *RESULTS* session (line 485-486).

Figure 6B and S6: The overall effect of Gdown1 access to the nucleus is clear, but more could have been learned with a quantitative comparison of the relative loss of total Pol II, ser5P and ser2P. For example, if ser2 is reduced more than ser5 this could imply an effect on elongation. Probing a panel C blot with the appropriate antibodies could have addressed this point.

Thank you for this good suggestion and we agree that checking the changes of Ser5P and Ser2P would provide more valuable information. We did this experiment and the quality of the WB data was ok. Since the stable cell lines used in these assays were cell pools with the expression levels of the total and the nuclear GDOWN1 varied from cell to cell, checking the total protein levels by WB might underestimate the real expression changes. Please refer to our response to your related question #8 and the Response Figure 2A-B. In addition, we carried out the IF experiments using various antibodies and carefully quantified the results (Figure 6C and Figure 6—figure supplement 1B) and the data indicated that upon the expression of the GDOWN1(*10M*) and NLS- GDOWN1(*10M*) (on day 4 post Dox addition), both Ser5P and Ser2P were reduced upon the expression of the nuclear GDOWN1, suggesting that both transcription initiation and elongation were inhibited. However, the statistical analyses shown in Figure 6—figure supplement 1C (better presented in Response Figure 2C-D) indicated that the S5P signal dropped more severe than the S2P signal, suggesting that the transcriptional initiation was more sensitive to the regulation of GDOWN1. The point was mentioned in both the RESULTS session.

I could not understand the result in panel 6C. If nuclear export is blocked by LMB, how can most of HeLa Pol II be cytoplasmic after LMB treatment?

This is an interesting question! Although Pol II is not a target of CRM1 by itself, it has been known that its assembly factors, such as RPAP2 and RPAP4/GPN1 are cargoes of CRM1(Forget et al., 2010; Forget et al., 2013). Upon LMB treatment, these factors will be accumulated in the nucleus and result in the dramatic reduction of Pol II assembly and its cytoplasm retention. This piece of data is now moved to the Figure 6—figure supplement 1A.

Points in the Discussion:How is Gdown1 acting in the nucleus? The earlier biochemical work provided a number of possibilities but those aren't investigated here or extensively discussed. Is Gdown1 mostly acting to tie up Pol II so that it cannot make PICs, or is it also/mostly acting downstream of initiation to block elongation? As noted above, determining whether ser5 and ser2 Pol II are affected differently might have shed light on this. Additionally, it would have been useful to know if nuclear Gdown1 is chromatin associated (probably talking to elongation) or free in the nucleoplasm (probably bound to Pol II). The point about effects on elongation seems particularly important when trying to understand how nuclear Gdown1 reduced overall Pol II levels. In particular, if elongation were severely affected polymerases could be stalled for extended periods at difficult-to-traverse locations, provoking the ubiquitination and clearance of those complexes (and thus reducing overall Pol II levels).

Very good questions and suggestions, thank you! Our original goal was to dissect GDOWN1’s functional mechanisms in regulating transcription in the nucleus. However, the fact that it did not enter the nucleus redirected us to a new route of digging the molecular basis of its cytoplasmic retention. It was exciting to uncover that the native GDOWN1 was translocated into the nucleus upon the oxidative stresses, possibly opening a new avenue of identifying its physiological roles in the cellular adaptation to stresses.

When being translocated into the nucleus, GDOWN1 triggered the global repression of transcription. The IF signals of the un-phosphorylated Pol II (preferably detected by the Pol II-CTD antibody 8WG16, Abcam, ab817; please also refer to our response to Reviewer 2’s Q8 for some evidence about this antibody’s specificity), and the phosphorylated forms of Pol II on the Ser5 residues (detected by the Pol II-S5P-CTD antibody CTD4H8, Biolegend, 904001) and on Ser2 residues (detected by the Pol II-S2P-CTD antibody, Abcam, ab5095) were all reduced in the cells with the expression of the nuclear GDOWN1 at 4 days after the Dox induction (Figure 6B and Figure 6—figure supplement 1A).

To confirm these data, we carefully tested the changes of these various forms of Pol II via WB experiments using a series of Pol II antibodies (Author response image 2). A newly ordered Pol II-NTD antibody (clone D8L4Y, CST) was employed to more accurately detect the total level of Pol II, including the un-phosphorylated and the phosphorylated forms (Figure 6B). As seen in Author response image 2 and the quantified data in Author response image 2, the protein levels of the total, the un-phosphorylated, and the S5P form of Pol II all got reduced significantly upon the nuclear accumulation of GDOWN1 (the *10M* mutant VS the wild type), while the change of the S2P-Pol II was relatively small. We also ordered another Ser5P antibody, while unfortunately, after a 3 months-waiting for its oversea shipping, it turned out it did not work well.

**Author response image 2. sa2fig2:** The nuclear accumulation of GDOWN1 causes reduction of Pol II. (A) Using four different Pol II antibodies to detect the changes of Pol II in the indicated cell lines upon the expression of GDOWN1 on day 4 post Dox addition; [T: total, the whole cell lysate; C: cytosol; N: nuclei] (B) The quantification of the data shown in (A); (C) The nuclear accumulation of GDOWN1 in the iNLS-GDOWN1(*10M*) cells causes differential decreases of the distinct forms of Pol II in IF assays; (D) The quantification of the data shown in (C).

To confirm the change of S2P-Pol II shown in Figure 6—figure supplement 1A [Anti-RPB1 CTD (S2P), Abcam, ab5095], we repeated the IF experiments with another S2P specific antibody [Anti-RPB1 CTD (S2P), Abcam, clone3E10]. As seen in Author response image 2 (also contain a subset of images in Figure 6B for comparison), both S2P antibodies generated very similar results that the levels of the S2P-Pol II went down in the cells expressing the nuclear GDOWN1. However, the extent of the reduction for S2P was much less than that of the S5P. We recently noticed that at early time points, accumulation of the nuclear GDOWN1 seemed to primarily affect Ser5P levels (data not shown).

We also confirmed that the loss of Pol II signals was attributed to protease-dependent protein degradation (-/+ MG132 treatment, data not shown), while the mechanisms behind remain to be uncovered. We added some of the above information in the *DISCUSSION* session.

All of our current experimental evidence supports the hypothesis that once entering the nucleus, GDOWN1 preferentially binds to and affects the unphosphorylated Pol II and the S5P-Pol II engaged in the initiation stage, and causes their degradation. At later stage, its long-lasting accumulation in the nucleus also leads to the severe reduction of S2P-Pol II. More thorough analyses are definitely necessary to dissect the mechanistic details of these processes, and we are working on it to make it clear.

Finally, lines 604-607: The authors assert that EU labeling in nucleoli is drastically reduced upon Gdown1 entry into nuclei, showing that Gdown1 interferes with both Pol I and Pol III activity. There is no quantitation provided for the assertion that nucleolar labeling is reduced, but regardless such reduction would not show a direct effect on Pol I (or Pol III) activity- strong reductions in mRNA synthesis should feed back into reductions in rRNA synthesis.

Thank you for helping us find this inaccurate statement in our manuscript. Actually, it was shown in the literature that the transcription rate of Pol II was much higher than Pol I and Pol III (Jackson et al., 2000), and ~80% of the EU labeled transcripts via a short pulse (within one hour) were Pol II products (Nozawa et al., 2017). Therefore, we partially missed the detection about the dynamic changes of Pol I and Pol III transcripts in our 20 min-EU pulse labeling assays, and the major signals and the associated changes shown in our IF results were mainly representing Pol II transcripts. Yes, you are right, the reduction of the nucleolar labeling might be directly or indirectly affected by GDOWN1 and the associated Pol II activity, while we are currently lacking of evidence. Based on the above considerations, we removed this statement from the *DISCUSSION* session.

Reviewer #2 (Recommendations for the authors):We have a few suggestions to increase the clarity of this manuscript:1. The current title is a bit complicated. Can the authors make the title simpler?

Thank you for your suggestion! We simplified this title as followed, “Overcoming cytoplasmic retention of GDOWN1 modulates global transcription and facilitates adaptation to cellular stresses”.

2. The authors have some factually incorrect statements in their introduction. They should try to improve the content and presentation of the introduction. Examples:a. In the introduction (line 57-64), they discuss in vitro work done on GDOWN1, but they do not mention the 2018 structure from Roeder (doi: 10.1038/s41594-018-0118-5). Can the authors please add this paper, as it also clarifies what is known about GDOWN1-Pol II interactions?b. In the introduction (line 89), they mention ChIP-seq data of Gdown1 and elongating Pol II. Can they please clarify this sentence? It is confusing to understand.c. Line 38: RNA polymerase II is in fact not the only polymerase that synthesis protein coding genes in eukaryotes (e.g., mitochondrial RNA polymerase also synthesizes protein coding mRNA).

Thank you for the Reviewer’s comments about the *INTRODUCTION* session. We carefully revised this part and corrected or clarified all the points mentioned by the reviewer and some others.

a. Professor Roeder’s paper in 2018 really is a very important paper demonstrating that the GDOWN1/Pol II interactions via the structural point of view, and further providing evidence about the associated effects on transcription. We now added this reference in the *INTRODUCTION* session and pointed out its main findings. b. We clarified this point in the revised manuscript.

c. This mistake was corrected.

3. The figures are very complicated with many panels. Could the authors consider making more subpanels? For example, Figure 1D is showing two different things; what factors GDOWN1 interacts with, and then various factors interacting with other factors. We recommend separating these figures to increase the clarity of the paper.

Thank you for the Reviewer’s suggestion! We now split Figure 1D into two panels, D and E. All the other main figures are also carefully checked. We agree that they are complicated, while we find logically the current organization seems to be reasonable. Therefore, we would like to keep them as they are. If the Reviewers do think some of the other figures need to be further split or they are organized inappropriately, please specify the figure number(s) and we would like to try our best to improve, thank you!

4. We have a few suggestions for improving Figure 4A. It is of particular interest to see how NES1, NES2, and then the double NES mutation influences nuclear levels. Could they more directly compare these three mutation-types by putting quantitative analysis of all constructs next to each other?

We thank the Reviewer for this good suggestion! We now reorganized the quantitative data analyses of all the constructs shown in Figure 4A and show them side by side for better comparison (Figure 4B) and the detailed description was added in the *RESULTS* Session. Upon LMB treatment, the N/C ratios of the three GDOWN1 mutants are ordered as mCAS (g) > mNES2 (e) >> mNES1 (c), which indicates that the CRM1-independent cytoplasmic localization of GDOWN1 is mainly determined by CAS, while NES2 also plays a role on this regulation. In the presence of LMB, there is no significant difference between mNES2 (e) and mNES1+2 (f), indicating that NES1 is not involved in regulating GDOWN1’s localization via the CRM1-independent manner. The significantly enhanced N/C ratio of NES1+CAS and NES2+CAS double mutants comparing to the NES single mutants (d to c, b to e) in the presence of LMB further confirms that CAS is the core motif in controlling GDOWN1’s cytoplasmic localization.

5. While the evolutionary analysis in Figure 4B+C is exciting, we feel it takes away from the figures purpose. We suggest moving figure 4B+C to the supplementals and then expanding on the analysis of the different NES in the main figure.

Thank you for this suggestion! We reorganized the data arrangement in Figure 4 and further expanded the data analyses of various GDOWN1 mutants (Figure 4B). We agree that the part of the evolutionary analysis is slightly far away from the main route of the topic, while these important observations provide a strong support for the conservation of these key amino acids in controlling the subcellular localization of Gdown1 during evolution. Logically, it is suitable to be in this main figure. Therefore, we would like to keep it as it is to emphasize the importance of these amino acids.

6. Please make a cartoon describing that 10M is a triple mutation of all three identified domains in figure 5. We generally to produce an additional supplementary figure panel that summarizes all used constructs in a graphical representation.

It’s a good suggestion. We have added a panel of cartoons on the top of Figure 5B. In addition, we summarized all the constructs in a table of the *MATERIALS AND METHODS* session.

7. In Figure 6, the authors provide evidence for a decrease in EU RNA incorporation in the context of the triple mutant. It would strengthen the manuscript if the authors could demonstrate a transcriptional defect by providing data from a complementary technique.

Thank you for your suggestion! We agree that providing more evidence in addition to the EU staining would be better. In this case, when global transcription is dramatically reduced, technically it is challenging to quantify the changes in RT-qPCR assays or RNA-Seq analyses. We now provide some normalized RT-qPCR data in Author response image 3. The total RNAs were extracted from equal numbers of the four cell lines shown in Figure 6A on day 3 after the mock treatment or Dox incubation, respectively. We found that the overall quantities of the total RNA were generally not changed, suggesting that the bulk of cellular rRNAs may not be significantly affected, at least on the third day after the nuclear GDOWN1 expression (data not shown). In order to accurately quantify the expression changes in qPCR assays, we introduced the total RNA from *Arabidopsis thaliana* as an internal reference. The reverse transcription reactions were performed using the mixed samples composed of 700 ng of the total RNA from each type of HeLa stable cell line (-/+Dox) and 300 ng of the total RNA from Arabidopsis. The resultant cDNA samples were further used as templates in qPCR reactions and an Arabidopsis unique amplicon (in *Actin2* gene) was used as an “internal control” for data quantification (primers used are listed in Response Table 1).

**Author response image 3. sa2fig3:** The changes of the indicated transcripts after 3 days of induction in cells expressing the indicated GDOWN1 variants. Total RNA samples from HeLa cells expressing GDOWN1 (or its mutant) for mock or 3 days of Dox induction were extracted. 700 ng RNA from the indicated cell lines and 300 ng RNA from Arabidopsis were mixed and used to do reverse transcription. The cDNA samples were further prepared and used in qPCR assays and the Actin2 gene of *Arabidopsis* was used as a reference gene for data normalization.

The results shown that the levels of the highly expressed mRNA species, such as *GAPDH* and *H2B*, were significantly reduced in the two cell lines expressing the nuclear-localized GDOWN1(*10M*) mutants, indicating that the nuclear GDOWN1 may play an important role in repressing Pol II-catalyzed global transcription of mRNAs (and some lncRNAs). The level of the 28S rRNA didn’t change much after 3 days of induction in the cells expressing GDOWN1, *10M* or NLS-GDOWN1, but reduced in the cells expressing NLS^-1^*0M* to some extent, which might be correlated to the strong reduction of mRNA synthesis. Taken together, these results and the data shown in Figure 6 demonstrate that the massive nuclear accumulation of GDOWN1 results in a dramatic reduction of the global transcription of Pol II.

**Author response table 1. sa2table1:** The sequences of primers used in RT-qPCR assays.

qPCR Primers	Sequences
AtActin2-F	TGTGCCAATCTAGGAGGGTTT
AtActin2-R	TTTCCCGCTCTGCTGTTGT
hGAPDH-F	GTCAGCCGCATCTTCTTTTG
hGAPDH-R	GCGCCCAATACGACCAAATC
h28S rRNA-F	GACCCGAAAGATGGTGAACTATG
h28S rRNA-R	CGATTTGCACGTCAGGACCG
hH2B-F	CCTGAGCCAGCCAAGTCTGC
hH2B-R	TGCGGCTGCGCTTGCGCTTC

8. Could the authors provide additional quantification of Figure S6B?

We guess the Reviewer meant ‘Figure 6C’ since there was no ‘Figure S6C’ in our original manuscript submitted. However, the bar graph shown on the right in Figure 6C is the quantitative analyses of WB result. We think the error bars are not essential for the quantification analyses of the WB results.

In Figure 6C of our first-time submitted manuscript, we used the Pol II CTD antibody [clone 8WG16, (Abcam)] to indicate the changes of the total Pol II, while the specificity of this antibody was somewhat controversial. Previously, we actually confirmed in our in vitro kinase assays, this antibody preferentially recognized the unphosphorylated Pol II (only one major band shown in the WB data), although it could also recognize the S2P-Pol II when it became dominant in the sample{Cheng, 2007 #5106}. We borrowed that piece of data (the supplemental figure 2 in that *JBC*paper) and provided it in Response Figure 4A.. Thus, we made some corrections in the revised manuscript and considered it as the antibody specifically recognizes the un-phosphorylated Pol II. The WB experiments using this antibody were repeated and very similar results were obtained (both sets of data are shown in parallel in Author response image 4). To simplify the data for publication, we only provided one set in Figure 6—figure supplement 1A in the revised manuscript.

**Author response image 4. sa2fig4:** The changes of the level of the un-phosphorylated RPB1 and its nuclear fraction upon the nuclear accumulation of GDOWN1 or the treatment of LMB. A. The target specificity test of the Pol II-CTD antibody, 8WG16. in vitro kinase assays were carried out using the purified P-TEFb as the kinase, and either the purified factors or the HeLa nuclear extract (HNE) as the substrates. The 8WG16 antibody was employed for Pol II detection in the WB experiment. This data was published as the Supplementary Figure 2 in Cheng et al., *JBC*, 2007, 282(30): 21901-21912. B-E. The indicated cell lines were cultured and subjected to cell fractionation and WB analyses. Two independent experiments were performed and the results are shown in B&C, and D&E, respectively. Antibodies used for detection were labeled on the left side. The relative changes of the RPB1 level as a whole and the ratio of its nuclear fraction were shown in the bar graphs on the right. As a very long-life protein, histone H3 was used as the internal control for data normalization. TUB: TUBULIN; the nuclear fraction (N), and the total cell lysate (T) .

To improve the accuracy and reliability of our result for the total Pol II detection, we employed a new Pol II NTD antibody [Anti-RPB1 NTD, clone D8L4Y, (CST)], and the result also demonstrated that the level of Pol II was significantly decreased when the nuclear GDOWN1 was expressed. The original Figure 6C was thus replaced with this new set of data, and became the new Figure 6B in the revised manuscript.

9. The authors should also provide error bars for Figure S6C.

Figure S6B in the previously submitted manuscript demonstrated the controls for

the WB experiment shown in the main Figure 6 to support the effectiveness and purity of our cell fractionation experiment. Both TUBULIN and FBL were used as the reference proteins to represent the cytoplasmic protein and the nuclear protein, respectively, and the purpose of showing these controls was mainly for the qualitative use, but not for the quantitative use. We understand why the Reviewer ask for data quantification here since the level of these controls (especially FBL) apparently reduced quite a lot upon the expression of the nuclear GDOWN1. Because GDOWN1 causes a global transcriptional repression, the effect on the level of each protein actually varies based on the half time of the protein. Therefore, for data quantification, we refer to use histone 3 because histones proteins are the well-known, most stable proteins in cells.

10. As the authors are well aware, Pol II antibodies show sub-optimal specificity. Could the authors provide data from additional Pol II antibodies (e.g., S2P, S5P, RPB3) for Figure S6C?

We thank the reviewer for this good suggestion! We ordered several new Pol II antibodies and provided some new IF and WB data. Please refer our responses to Reviewer 1’s Q8 and to Reviewer 2’s Q8. Overall the results were consistent with our previous data, and the data quantification suggest that the unphospho-Pol II and the S5P-Pol II were more severely and rapidly affected than the S2P-Pol II.

11. In lines 455-456, the authors mention in vitro transcription assays, but do not cite anything. Please cite the study being referred to.

Thanks for your reminding! We cited two corresponding references (Cheng et al., 2012; Jishage et al., 2012).

12. We ask the authors to discuss why GDOWN1 apparently interacts with so many transcriptional regulators in the cytoplasm. Is GDOWN1 regulating these factors by interacting with them in the cytoplasm?

Very good point! We checked the subcellular localization of some transcriptional regulators, such as NELF-E etc., and found they were only present in the nucleus. Thus, the BiFC signals about the ‘GDOWN1•transcriptional regulator’ interactions may only represent their interaction potential in the nucleus instead of the actual occurrence in the cytoplasm between their endogenous counterparts. We hypothesize that under the physiological conditions, GDOWN1 may be transiently and unstably interacting to the newly translated transcriptional regulators in the cytoplasm while it does NOT trap them in the cytoplasm or affect their functions. When both factors are overexpressed in cells in our BiFC assays, GDOWN1 is able to trap a fraction of its nuclear partners in the cytoplasm and the transient interaction can be stabilized by BiFC (the BiFC signal is irreversible because of the covalent bonds set up in between the two split parts of a fluorescent protein upon the BiFC interaction occurs). On the other hand, from the mass spectrometry data and some of our GST-pull down or IP data, we confirmed that GDOWN1 interacted to Pol II components and their assembly factors in the cytoplasm. In addition, we observed that in *GDOWN1 KO* cells, the nuclear/cytoplasmic ratio of RPB1 decreased (data not shown). Therefore, GDOWN1 may contribute to the assembly and also the nuclear import of Pol II. More thorough analyses are needed to dissect its roles in these processes.

13. The Alphafold structure in supplement figure 4 – 1A is difficult to see. Can the authors increase clarity by changing to a white background and focusing more on the regions they are showing? Please also add the same cartoon that is used throughout the paper to further help clarifications.

Thank you for your suggestion! We have changed the background to white and focused more on the two NESs and the CAS motif. We also added a cartoon to further facilitate clarification according to your advice. The corresponding changes have been made in Figure 4—figure supplement 1A.

14. The authors should also carefully check their manuscript for missing prepositions etc.

Thank you for your kind reminder! We carefully checked and corrected the grammatical mistakes throughout our manuscript.

References

Cheng, B., Li, T., Rahl, P.B., Adamson, T.E., Loudas, N.B., Guo, J., Varzavand, K., Cooper, J.J., Hu, X., Gnatt, A.,etal.(2012). Functional association of Gdown1 with RNA polymerase II poised on human genes. Mol Cell 45, 38-50. Doi: http://doi.org/10.1016/j.molcel.2011.10.022, Accession number: 22244331

Forget, D., Lacombe, A.A., Cloutier, P., Al-Khoury, R., Bouchard, A., Lavallee-Adam, M., Faubert, D., Jeronimo, C., Blanchette, M., and Coulombe, B. (2010). The protein interaction network of the human transcription machinery reveals a role for the conserved GTPase RPAP4/GPN1 and microtubule assembly in nuclear import and biogenesis of RNA polymerase II. Mol Cell Proteomics 9, 2827-2839. Doi: http://doi.org/10.1074/mcp.M110.003616, Accession number: 20855544

Forget, D., Lacombe, A.A., Cloutier, P., Lavallee-Adam, M., Blanchette, M., and Coulombe, B. (2013). Nuclear import of RNA polymerase II is coupled with nucleocytoplasmic shuttling of the RNA polymerase II-associated protein 2. Nucleic Acids Res 41, 6881-6891. Doi: http://doi.org/10.1093/nar/gkt455, Accession number: 23723243

Hu, X., Malik, S., Negroiu, C.C., Hubbard, K., Velalar, C.N., Hampton, B., Grosu, D., Catalano, J., Roeder, R.G., and Gnatt, A. (2006). A Mediator-responsive form of metazoan RNA polymerase II. Proc Natl Acad Sci U S A 103, 9506-9511. Doi: http://doi.org/10.1073/pnas.0603702103, Accession number: 16769904

Jackson, D.A., Pombo, A., and Iborra, F. (2000). The balance sheet for transcription: an analysis of nuclear RNA metabolism in mammalian cells. FASEB J 14, 242-254. Doi: Accession number: 10657981

Jishage, M., Malik, S., Wagner, U., Uberheide, B., Ishihama, Y., Hu, X., Chait, B.T., Gnatt, A., Ren, B., and Roeder, R.G. (2012). Transcriptional regulation by Pol II(G) involving mediator and competitive interactions of Gdown1 and TFIIF with Pol II. Mol Cell 45, 51-63. Doi: http://doi.org/10.1016/j.molcel.2011.12.014, Accession number: 22244332

Jishage, M., Yu, X., Shi, Y., Ganesan, S.J., Chen, W.Y., Sali, A., Chait, B.T., Asturias, F.J., and Roeder, R.G. (2018). Architecture of Pol II(G) and molecular mechanism of transcription regulation by Gdown1. Nat Struct Mol Biol 25, 859-867. Doi:

http://doi.org/10.1038/s41594-018-0118-5, Accession number: 30190596

Nozawa, R.S., Boteva, L., Soares, D.C., Naughton, C., Dun, A.R., Buckle, A., Ramsahoye, B., Bruton, P.C., Saleeb, R.S., Arnedo, M.,etal. (2017). SAF-A Regulates Interphase Chromosome Structure through Oligomerization with Chromatin-Associated RNAs. Cell 169, 1214-1227 e1218. Doi: http://doi.org/10.1016/j.cell.2017.05.029, Accession number: 28622508